# Adversarial Robustness through Random Weight Sampling

**Yanxiang Ma, Minjing Dong, Chang Xu**[*]
School of Computer Science, University of Sydney
{yama9404, mdon0736}@uni.sydney.edu.au
c.xu@sydney.edu.au

## Abstract

Deep neural networks have been found to be vulnerable in a variety of tasks. Adversarial attacks can manipulate network outputs, resulting in incorrect predictions. Adversarial defense methods aim to improve the adversarial robustness of networks by countering potential attacks. In addition to traditional defense approaches, randomized defense mechanisms have recently received increasing attention from researchers. These methods introduce different types of perturbations during the inference phase to destabilize adversarial attacks. Although promising empirical results have been demonstrated by these approaches, the defense performance is quite sensitive to the randomness parameters, which are always manually tuned without further analysis. On the contrary, we propose incorporating random weights into the optimization to exploit the potential of randomized defense fully. To perform better optimization of randomness parameters, we conduct a theoretical analysis of the connections between randomness parameters and gradient similarity as well as natural performance. From these two aspects, we suggest imposing theoretically-guided constraints on random weights during optimizations, as these weights play a critical role in balancing natural performance and adversarial robustness. We derive both the upper and lower bounds of random weight parameters by considering prediction bias and gradient similarity. In this study, we introduce the Constrained Trainable Random Weight (CTRW), which adds random weight parameters to the optimization and includes a constraint guided by the upper and lower bounds to achieve better trade-offs between natural and robust accuracy. We evaluate the effectiveness of CTRW on several datasets and benchmark convolutional neural networks. Our results indicate that our model achieves a robust accuracy approximately 16% to 17% higher than the baseline model under PGD-20 and 22% to 25% higher on Auto Attack.

## 1 Introduction

Deep neural networks (DNNs) have rapidly become valuable tools for processing complex information [1, 2, 3, 4, 5]. However, their widespread usage has also increased the risk of adversarial attacks [6], making it essential to design robust learning techniques that can withstand such attacks [7]. Many works have demonstrated that DNNs are highly sensitive to small changes in input, which has led to the development of adversarial attack algorithms such as the Fast Gradient Sign Method (FGSM) [8], Projected Gradient Descent (PGD) [9] based attack, and Automatic Attack (AA) [10], among others. These white-box attacks assume that the network structure and parameters are accessible at the time of the attack, whereas they are black-box otherwise. Among them, white-box attacks are always regarded as critical threats to DNNs since the gradients are available for attack generation.

---

[*]Corresponding author.

Various defense methods, such as PGD-based Adversarial Training (AT) [9], have provided the basic algorithm for training robust models against adversarial attacks. Data preprocessing techniques are also used to enhance model adversarial robustness [11]. Besides these defense mechanisms, randomized techniques have gained wide interest in improving adversarial robustness since these white-box attacks are very sensitive to gradients to some extent [12]. Increasing the randomness of the network can reduce the similarity of gradients after each inference, which ultimately lowers the success rate of attack algorithms. Recent effective adversarial defense efforts have added randomness to specific modules or intermediate results of the network, demonstrating superior performance [7, 13, 14, 15, 16].

However, existing randomized defense algorithms always treat randomness parameters such as $\mu$ and $\sigma$ as hyperparameters [7, 13, 15], requiring significant tuning experiments. For instance, Li et al. [15] have set the mean value as 0 and the standard deviation as 1, following the standard Gaussian distribution. Although manually tuning the randomness parameters can be effective since the defense performance is sensitive to these hyperparameters, the correlation between randomness parameters and adversarial robustness is still unexplored. Thus, instead of manually tuning the randomness parameters through methods like grid search, we propose discovering theoretically-guided randomness parameters that can achieve better adversarial robustness. Specifically, we focus on the gradients in randomized mechanisms and the prediction bias when randomness is involved. By exploring the constraints on the magnitude of randomness, we establish connections between randomness parameters and gradient similarity as well as natural performance.

Based on previous studies, we propose a defense module called Constrained Trainable Random Weight (CTRW). We update the random parameters, including $\mu$ and $\sigma$, by training them in parallel with the neural network to ensure that the block can adapt to DNNs with different structures. Furthermore, we constrain the variance according to the established constraints after each update, ensuring that the variance always remains within the optimal range. In this work, we conducted sufficient evaluations to demonstrate the superiority of CTRW on different benchmarks. Moreover, we provide empirical evidence for the theoretical analysis by conducting experiments on the location of CTRW and the constraints. In our experiments under Automatic Attack (AA) on ResNet18 [17], the model with CTRW improved the robust accuracy by $25.87\%$ over the baseline model. On the Wide Residual Network (WRN) [18], the module improved the robustness accuracy by $21.99\%$ over the baseline and by $6.47\%$ over the state-of-the-art (SOTA) random algorithm.

## 2 Randomized Parameter Optimization with Constraints

Let $\mathbb{H}$ be a hypothesis space of classification task and we consider a classifier $h \in \mathbb{H}$ which maps the input $x$ to the logits $h(x)$. Adversarial attacks are designed to maximize the loss function $\mathcal{L}$ with a small perturbation $\Delta$. Given the ground truth label $y$ and maximum perturbation size $\epsilon_d$, the adversarial examples $\tilde{x} = x + \Delta$ can be generated via gradient ascent in white-box setting as

$$\tilde{x}^* = \underset{\tilde{x}:\|\tilde{x}-x\|_p \leq \epsilon_d}{\operatorname{argmax}} \mathcal{L}(h(\tilde{x}), y), \tag{1}$$

where the perturbation $\Delta$ is constrained by $l_p$-norm. Traditional adversarial training algorithm performs a min-max game as

$$h^* = \underset{h \in \mathbb{H}}{\operatorname{argmin}} \mathbb{E}_{x,y \sim X,Y}[\mathcal{L}(h(\tilde{x}), y], \tag{2}$$

To further improve the adversarial robustness, randomized mechanisms are introduced to prevent attackers from deriving precise and powerful adversarial examples via the involvement of randomness during attacking and inference phases [7, 13, 14, 19]. Given two classifier $h_1$ and $h_2$ sampled from a subset $\mathcal{H} \subset \mathbb{H}$, the defense scheme can be formulated as

$$\mathcal{H}^* = \underset{h_2 \in \mathcal{H}}{\operatorname{argmin}} \mathbb{E}_{x,y \sim X,Y}[\mathcal{L}(h_2(\tilde{x}_1'), y], \text{ where } \tilde{x}_1' = \underset{\tilde{x}_1':\|\tilde{x}_1'-x\|_p \leq \epsilon_d}{\operatorname{argmax}} \mathcal{L}(h_1(\tilde{x}_1'), y). \tag{3}$$

Note that sampled classifiers from $\mathcal{H}$ share all the weight parameters except for the injected noise. Thus, without additional cost, different classifiers can be randomly sampled from optimized $\mathcal{H}^*$ during each forwarding to form a defense. The white-box attack is transferred to a black-box attack when the adversarial example $\tilde{x}$ generated on $h_1$ is difficult to generalize to another sampled classifier $h_2$.

In this work, we investigate the incorporation of a random weight during the network forwarding process, whereby layers $h_1$ and $h_2$ share all the weight parameters except for the random weight.

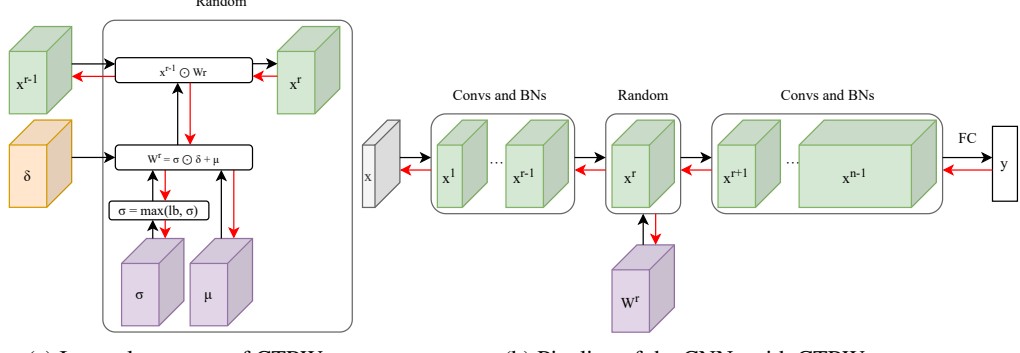

(a) Internal structure of CTRW.  (b) Pipeline of the CNNs with CTRW.

Figure 1: CTRW and the whole random adversarial defense framework pipeline.

To be specific, let $r$ be the layer that employs the random weight, and let $W^r$ be the weight in the $r$-th layer. We assume that $W^r$ follows a normal distribution $N(\mu, \sigma^2)$, where $\mu$ and $\sigma^2$ denote the mean and variance, respectively. To ensure that the random weight is adapted to the training data, we involve $\mu$ and $\sigma$ in the network optimization process. Although the random sampling process is not differentiable, the parameters can be optimized via a reparameterization trick as

$$W^r = \mu + \delta \odot \sigma, \tag{4}$$

where $\delta$ is sampled from a standard Gaussian distribution, *i.e.* $N(0,1)$. In $h1$ and $h2$, the only difference lies in $\delta$, which results in different $W^r$ values. By utilizing the technique mentioned in Equation 4, the training process of the random parameters can be visualized as shown in Figure 1a, using the pipeline depicted in Figure 1b. The variance $\sigma^2$ plays a crucial role in determining the level of randomness as it directly controls its magnitude. To ensure the effectiveness of randomized mechanisms, a relatively large $\sigma$ is required to guarantee the difficulty of transferring adversarial examples within the classifier subset $\mathcal{H}$. However, an excessively large $\sigma$ can significantly impact the natural performance. While a naive solution for optimizing $\sigma$ could involve adversarial training on both clean examples $x$ and adversarial examples $\tilde{x}$, such as Trades [11], it incurs additional training costs on clean examples. To achieve a unified optimization of $\sigma$, we propose an implicit approach that imposes constraints with upper and lower bounds on $\sigma$ to ensure both the natural performance and defense capability. Let $W[i]$ be the weight including $W^r$ of $h_i$, the objective function can be redefined as

$$\mathcal{H}^* = \underset{h_{W[2]} \in \mathcal{H}}{\operatorname{argmin}} \mathbb{E}_{x,y \sim X,Y}[\mathcal{L}(h_{W[2]}(\tilde{x}), y], \text{ where } \tilde{x} = \underset{\tilde{x}:\|\tilde{x}-x\|_p \leq \epsilon_d}{\operatorname{argmax}} \mathcal{L}(h_{W[1]}(\tilde{x}), y),$$
$$\text{s.t. } \sum \sigma \in [A, B], \tag{5}$$

where $\sum \sigma$ denotes the sum of every elements in $\sigma$, $A$ and $B$ represent the lower and upper bounds of $\sigma$, respectively. Therefore, when $\sigma$ is too low, the associated randomness may be insufficient, making the adversarial example $\tilde{x}$ easily transfer between $h1$ and $h2$. On the other hand, if $\sigma$ is too high, excessive randomness can lead to performance degradation for any $h \in \mathcal{H}$. To understand the relationship between $A$, $B$, and the optimization objective, selecting them through grid search is not sufficient. Thus, to better tune the bounds of $\sigma$, we aim to establish, from a theoretical perspective, how $A$ and $B$ impact the gap between $h1$ and $h2$, as well as the performance degradation of any $h \in \mathcal{H}$. The theoretical analysis will guide for tuning the reference. Therefore, we have developed objectives for the upper and lower bounds, respectively.

## 2.1 Lower Bound for Gradient Similarity

The white-box attacks such as FGSM, PGD, and MIFGSM [8, 9, 20] have high success rates in white-box settings due to the direct derivation of gradients in Eq. 1. However, in randomized mechanisms, the gradients change for the same input due to the random sampling of the classifier $h$ from $\mathcal{H}$ during each forwarding. The success of randomized defense can be attributed to the differences in gradients, resulting in poor perturbation generation during attacks and possible ineffectiveness in inference due

to the changes in output. To quantify and analyze the relationship between the random parameter and gradient difference, we propose measuring their cosine similarity. Formally, For any $h_1$ and $h_2$, the cosine similarity between gradient $\nabla$ on $x$ at different weights is computed as

$$cos(\nabla(x, W^r[1]), \nabla(x, W^r[2])). \tag{6}$$

The random weights are obtained through element-wise multiplication (Hadamard product) of the input of the layer in the neural network. To ensure that a large gradient difference, we require the cosine similarity of the gradients to be always less than a threshold $\epsilon_\nabla$. However, since the random weights are sampled from an unbounded distribution, we cannot guarantee that the cosine similarity will always meet this requirement, and can only provide a probabilistic bound. Assume that the random sampled weights are $W^r[1]$ and $W^r[2]$ respectively, the objective function can be formulated as

$$\mu, \sigma = \underset{W^r \sim N(\mu, \sigma^2)}{argmax} \ P\Big(cos(\nabla(x, W^r[1]), \nabla(x, W^r[2])) \leqslant \epsilon_\nabla\Big). \tag{7}$$

Considering the random weights on the $r$-th layer, there exists a constant mapping between the gradient of $x^{r-1}$ and the gradient of $x^0$ since the network parameters are all fixed. Thus, we focus on the gradient of $x^{r-1}$ instead of the one of $x^0$. Given random weights $W^r[1]$ and $W^r[2]$, we denote the outputs as $y[1]$ and $y[2]$ respectively. According to the chain rule, the gradient can be computed as the product of the gradient of the layer after $r$ and the gradient of the random layer. Thus, the angle $\langle\nabla(x, W^r[1]), \nabla(x, W^r[2])\rangle$ can be split into the sum of two angles, and then using the summation formula for the cosine function it can be derived that

$$\begin{aligned} cos(\nabla(x, W^r[1]), \nabla(x, W^r[2])) &= cos(\nabla^{r,r+1}[1], \nabla^{r,r+1}[2])cos(\nabla^{r+1,n}[1], \nabla^{r+1,n}[2]) \\ &\quad - sin(\nabla^{r,r+1}[1], \nabla^{r,r+1}[2])sin(\nabla^{r+1,n}[1], \nabla^{r+1,n}[2]), \end{aligned} \tag{8}$$

where $\nabla^{i,j}[1]$ and $\nabla^{i,j}[2]$ are the gradient of $x^i$ propagated from $x^j$ with $W[1]$ and $W[2]$ respectively. Assume that $y[1] - y[2] \leqslant \epsilon_y \to 0^+$, i.e. the angle $\langle y[1], y[2]\rangle \leqslant \epsilon_{ang}^y \to 0^+$. Since $\nabla 1_{r+1,n}$ and $\nabla 2_{r+1,n}$ are only related to $y[1]$ and $y[2]$ in a multiplicative manner, the upper bound of angle $\langle\nabla^{r+1,n}[1], \nabla^{r+1,n}[2]\rangle$ can be limited as $\epsilon_{ang}^\nabla \to 0$. Based on Eq. 8, the cosine similarity is mainly influenced by and positively correlated with $cos(\nabla 1_{r,r+1}, \nabla 2_{r,r+1})$. Note that the random layers only include the Hadamard product of random weight and the input, the derivatives $\nabla^{r,r+1}[1]$ and $\nabla^{r,r+1}[2]$ are exactly the random weight $W^r[1], W^r[2]$. Thus, Eq. 7 can be reformulated as

$$\mu, \sigma = \underset{\mu, \sigma \geqslant 0}{argmax} P(cos(W^r[1], W^r[2]) \leqslant \epsilon_r), \text{ where } W^r \sim N(\mu, \sigma^2), \tag{9}$$

where $\epsilon_r$ is the upper bound of the cosine similarity between $W^r[1]$ and $W^r[2]$, and $\epsilon_r \to 0^+$. Since each element in $W^r$ obeys a Gaussian distribution, we consider using the probability cumulative function of the Gaussian distribution to constrain the probabilities in Eq. 9.

Specifically, to ensure that $P(cos(W^r[1], W^r[2]) \leqslant \epsilon_r)$ is sufficiently large, a lower bound is devised in the form of a cumulative probability function of the standard normal distribution. This lower bound on the probability is specified as $F(\alpha)$. In order for $P(cos(W^r[1], W^r[2]) \leqslant \epsilon_r) < F(\alpha)$, the condition $\alpha < 0$ needs to be satisfied. Although $\alpha < 0$ seems to be a low-confidence guarantee, this bound plays an important role and its effectiveness will be empirically verified in Table 6. The proof of this part is in Eq. (10) in the first part of the supplementary material. Based on this, an upper bound on the $\sigma$ associated with the weights can be defined.

**Lemma 1.** *Consider a fixed $W^r[1]$ with size of $m$, and a constant $\epsilon_r$ that satisfies $\epsilon_r = o(1)$ and an $\alpha < 0$. Let $\epsilon_r' = \epsilon^r \times \|W^r[1]\|_p \|W^r[2]\|_p$, we have $\epsilon_r' = o(1)$. Assume that $\sum_{i=1}^m \mu_i \gg 0$. Let $F$ denote the cumulative function of a standard Gaussian distribution, the constraint $cos(W^r[1], W^r[2]) \leqslant \epsilon_r$ holds with the probability at least $F(\alpha)$, i.e. $P(cos(W^r[1], W^r[2]) \leqslant \epsilon_r) > F(\alpha)$, if*

$$\sum_{i=1}^m \sigma_i \geqslant \frac{\epsilon_r' - \sum_{i=1}^m \mu_i}{\alpha \sum_{i=1}^m |W^r[1]_i|} > 0, \tag{10}$$

*where $m$ is the size of $W^r$, and for any variation $a$, $a_i$ denotes the $i$th element of $a$.*

Due to Lemma1., when $\sigma$ has a high enough lower bound, it is guaranteed that $F(\alpha)$ has a large enough value such that probabilities in Eq. 9 are large enough. During training, $\sigma$ will be checked after each optimization and added by the average bias when $\sum \sigma$ is lower than the lower bound.

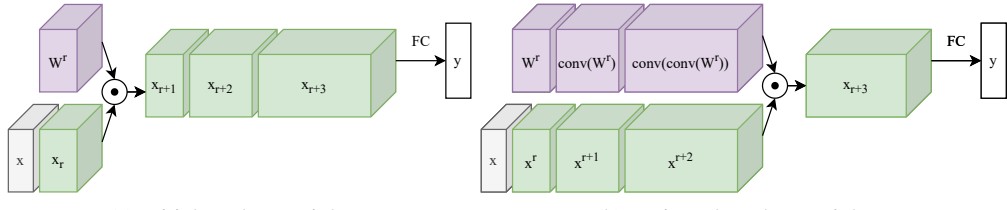

(a) Initial random weights.    (b) Projected random weights.

Figure 2: An Illustration of the projection of the random weight through different layers.

## 2.2 Upper Bound for Natural Performance

Section 2.1 highlights that a greater $\sigma$ leads to a higher likelihood of low similarity between gradients. However, a network's superior robust accuracy requires not only low cosine similarity between sampled networks but also satisfactory accuracy on clean examples. While a larger $\sigma$ may yield lower similarity between gradients, it may also result in low accuracy due to extreme randomization of network output. Hence, an upper bound needs to be imposed on $\sigma$ to ensure satisfactory accuracy. Additionally, the assumption in Section 2.1 regarding $|y1 - y2|$ requires high natural accuracy to hold. Both convolution and normalization being linear processes, random weights can be mapped to any layer before the Hadamard product calculation, as shown in Figure 2. After convolution, each random weight becomes a weighted sum of multiple Independent Identical Distribution (*i.i.d.*) Gaussian distribution, resulting in a Gaussian distribution. Let $\mathcal{H}_{a,b}$ denotes layer $a$ to $b$ in $\mathcal{H}$, $W^r_{i,j}$ denotes the $i, j$th element of $W^r$, and $W^l$ denotes weight in layer $l$, the mapping of $W^r$ between layer $l$ and layer $l-1$ can be can be formulated as

$$\mathcal{H}_{r+1,l}(W^r_{i,j}) = \mathcal{H}_{r+1,l-1}\Big( \sum_{k=-\frac{(ks-1)}{2}}^{\frac{(ks-1)}{2}} W^r_{i+k,j+l} * W^n_{i+k,j+l}\Big), \tag{11}$$

where $ks$ denotes the kernel size of the convolution layer. Let $\hat{W}^p$ denote the random weight projected to layer $p$, the specific form of the random weight after mapped by one convolution layer is

$$\mathcal{H}_{r+1,r+1}(\hat{W}^p_{i,j}) \sim N\Big( \sum_{k=-\frac{(ks-1)}{2}}^{\frac{(ks-1)}{2}} \mu_{i+k,j+l} * \hat{W}^{p+1}_{i+k,j+l}, \sum_{k=-\frac{(ks-1)}{2}}^{\frac{(ks-1)}{2}} (\sigma_{i+k,j+l} * \hat{W}^{p+1}_{i+k,j+l})^2\Big), \tag{12}$$

where $\mu_{i,j}$ and $\sigma_{i,j}$ denotes the $(i,j)$th element of $\mu$ and $\sigma$ respectively. The mapping of $W^r$ at layer $l$ is obtained by performing $l-r$ recursions on Eq. 11, and the specific form of $\mathcal{H}_{r+1,n}(W^r_{i,j})$ can be calculated be iterating Eq. 12 for $n-r$ times. The sum of each weighted example from the Gaussian distribution still obeys the Gaussian distribution. Denoting the standard representation of the Gaussian distribution in Eq. 12 as $\sigma'$, we can know that $\sigma \propto \sigma'$. Thus, $\sigma$ can be expressed by $\sigma'$.

**Lemma 2.** *Given an $n$ layer network and assuming ReLU to be the activation function, let the parameters of the random weights after mapping $n-r$ times be $\mu'$ and $\sigma'$, $x^l$ denotes the output feature map of layer $l$ and $W^l$ denotes the weights in layer $l$. The output difference $Y_1 - Y_2$ is hoped to be smaller than a minimal term $\epsilon_y$. The probability of $(Y_1 - Y_2) < \epsilon_y$ is defined as $P((Y_1 - Y_2) < \epsilon_y)$. And $P((Y_1 - Y_2) < \epsilon_y)$ is larger than $F(\frac{\sqrt{2}\beta}{2})$, i.e. $P((Y_1 - Y_2) < \epsilon_y) \geqslant F(\frac{\sqrt{2}\beta}{2})$, if $\sum_{i=1}^m \sigma'_i$ is upper bounded as,*

$$\sum_{i=1}^m \sigma'_i \leqslant \frac{\epsilon_y}{x^{n-1} \cdot W^n \beta}, \tag{13}$$

*where $m$ is the size of the feature map.*

The $\sigma'$ can be obtained through $\sigma$ according to the forward propagation process of the network. For a $\sigma$, the mapping of the upper bound mentioned in Eq.13 is for higher natural accuracy.

The upper bound $B$ in Eq. 5 can be defined as the one in Lemma 2., which provides a guarantee for natural performance. In contrast, the lower bound $A$ in Eq. 5 can be defined as the one in Lemma1.,

| **Algorithm 1** Adversarial Training under Constrain Using Black-Box Attack | **Algorithm 2** Adversarial Defense under White-Box Attack |
|---|---|
| **Input:** Training data:$X, Y$; Network:$h(x)$, with initialized weight:$W$, including $\mu$ and $\sigma$; Lower bound of $\sigma$:$\min_\sigma$; Attack iterations:$t$; Size of perturbation:$\epsilon_d$ | **Input:** Test data:$X, Y$; Network:$h(x)$ with pre-trained weight:$W$; Attack iterations:$t$; Size of perturbation:$\epsilon_d$ |
| | **Output:** $\tilde{y}$ |
| 1: Sample $\delta$ from a set of *i.i.d.*standard Gaussian distribution; | 1: **while** not converged **do** |
| 2: **while** not converged **do** | 2:   Sample a batch of $(x, y) \in (X, Y)$; |
| 3:   Sample a batch of $(x, y) \in (X, Y)$; | 3:   Re-sample $\delta$ from a set of *i.i.d.*standard Gaussian distribution as $\delta_1$; |
| 4:   Initialize adversarial perturbation $\Delta_x$; | 4:   Initialize adversarial perturbation $\Delta_x$; |
| 5:   **for** $i \leftarrow 1$ to $t$ **do** | 5:   **for** $i \leftarrow 1$ to $t$ **do** |
| 6:     $\Delta_x = clip_{\epsilon_d}(sign(\nabla_x \mathcal{L}(h(x, \delta), y)) \cdot \eta)$; | 6:     $\Delta_x = clip_{\epsilon_d}(\eta \cdot sign(\nabla_x \mathcal{L}(h(x + \Delta_x, \delta_1), y)))$ |
| 7:   **end for** | 7:   **end for** |
| 8:   Re-sample $\delta$ from a set of *i.i.d.*standard Gaussian distribution; | 8:   Re-sample $\delta$ from a set of *i.i.d.*standard Gaussian distribution as $\delta_2$; |
| 9:   $W = W - \nabla_W \mathcal{L}(h(x + \Delta_x, \delta), y)$; | 9:   $\tilde{y} = h(x + \Delta_x, \delta_2)$; |
| 10:   Constrain $\sigma$ by $\min_\sigma$; | 10: **end while** |
| 11: **end while** | 11: **return** $\tilde{y}$; |

which provides a guarantee for a low gradient similarity. Since robust accuracy is jointly influenced by gradient similarity and natural accuracy, A and B in Eq. 5 together guarantee a high robust accuracy. Thus, it is the $\sigma$ that is jointly constrained by A and B that keeps the randomness under control in such a way that robust accuracy is sufficiently high.

### 2.3   Adversarial Training and Defense

**Black-box Adversarial Training**   To train the whole pipeline, we use black-box attack as a part of adversarial training as the defense scheme in Eq.3. The optimization can be formulated as

$$\mathcal{H}^* = \underset{h_{W[2]} \in \mathcal{H}}{argmin} \, \mathbb{E}_{x,y \sim X,Y}[\mathcal{L}(h_{W[2]}(\tilde{x}), y], \text{ where } \tilde{x} = \underset{\tilde{x}:\|\tilde{x}-x\|_p \leq \epsilon_d}{argmax} \, \mathcal{L}(h_{W[1]}(\tilde{x}), y),$$

$$\text{s.t. } \sum_{i=1}^{m} \sigma_i \in [\frac{\epsilon'_r - \sum_{i=1}^{m} \mu_i}{\alpha \sum_{i=1}^{m} |W^r[1]_i|}, B], \text{ where } B \propto \frac{\epsilon_y}{x^{n-1} \cdot W^n \beta}. \tag{14}$$

It can be estimated that the lower bound is with a high probability less than $\frac{1}{|\alpha|}$ as a simplification of Eq. 10. Thus, based on the analysis in Section 2.2, the trade-off between natural accuracy and gradient similarity is controlled by $|\alpha|$. Having the lower bounds selected, $\sigma$ is constrained by it and is checked and transformed before each forward propagation period to ensure that $\sigma$ is in the bound.

**Adversarial Defense under White-box**   When inference, there are two different forward propagation processes. The attackers first conduct a set of forward propagation to derive the gradients for adversarial example generation. The second forward propagation will take the generated adversarial example back to the network for inference. However, due to the random sampling strategy in randomized techniques, random weight $W$ is resampled and thus performs a strong defense. The training and defense schemes of CTRW are demonstrated in Algorithm 1 and Algorithm 2.

## 3   Related Work

Deep Neural Networks (DNNs) are vulnerable to perturbations [21, 22], resulting in the development of numerous adversarial attacks and defense algorithms [7, 8, 9, 10, 13, 23, 24, 25, 26, 14, 15, 16, 27, 28, 20, 29, 30, 31]. One common approach for generating adversarial examples in attack algorithms is the use of gradient-based methods [8, 9, 10, 20]. Adversarial samples generated by adversarial attacks can be used to expand the diversity of the dataset[32]. However, adversarial samples also pose a problem of model robustness. In recent years, several stochastic adversarial defense algorithms have been proposed that leverage random sampling to alter the gradient of the model at each inference [7, 13, 14, 33, 15, 16]. Some of these methods utilize random numbers as weights that are multiplied

Table 1: The results of robust accuracy on CIFAR-10 of adversarial defense algorithms.

| Model | Method | Natural | PGD$^{20}$ | FGSM | AA | CW12 | MIFGSM | DeepFool |
|---|---|---|---|---|---|---|---|---|
| | baseline | 81.84 | 52.16 | 56.70 | 47.69 | 78.46 | 54.96 | 0.35 |
| ResNet18 | without constrain | 83.64 | 54.59 | 58.93 | 54.99 | 82.43 | 57.02 | 78.39 |
| | fixed std | 82.62 | 67.75 | 65.18 | 72.03 | 81.44 | 65.03 | 81.25 |
| | CTRW (ours) | **83.69** | **69.48** | **66.49** | **73.56** | **82.93** | **65.47** | **82.45** |
| | baseline | 85.27 | 55.06 | 60.65 | 52.24 | 82.24 | 58.47 | 0.40 |
| WRN34 | without constrain | 85.84 | 56.68 | 61.29 | 57.32 | 85.00 | 59.37 | 79.81 |
| | fixed std | 84.23 | 70.03 | 66.47 | 72.97 | 83.83 | 65.74 | 83.30 |
| | CTRW (ours) | **85.87** | **71.34** | **66.72** | **74.23** | **84.29** | **65.99** | **83.66** |

Table 2: The results of robust accuracy on CIFAR-100 of adversarial defense algorithms.

| Model | Method | Natural | PGD$^{20}$ | FGSM | AA | CW12 | MIFGSM | DeepFool |
|---|---|---|---|---|---|---|---|---|
| | baseline | 55.81 | 28.71 | 31.33 | 24.48 | 50.94 | 30.26 | 0.79 |
| ResNet18 | without constrain | **57.01** | 30.05 | 32.36 | 30.78 | 55.38 | 31.38 | 51.39 |
| | fixed std | 56.26 | 41.44 | 38.68 | 44.71 | 56.12 | 37.92 | **55.77** |
| | CTRW (ours) | 56.63 | **42.01** | **38.85** | **45.58** | **56.32** | **38.20** | 55.56 |
| | baseline | 60.11 | 31.69 | 35.40 | 28.36 | 57.11 | 34.14 | 0.23 |
| WRN34 | without constrain | 60.59 | 31.97 | 34.66 | 32.16 | 58.81 | 33.54 | 54.41 |
| | fixed std | 60.12 | 44.22 | 40.17 | 48.18 | **60.07** | 39.64 | **59.54** |
| | CTRW (ours) | **60.62** | **44.33** | **40.33** | **50.47** | 59.73 | **39.90** | 59.30 |

with intermediate results of the model [7, 13]. Additionally, random noise can be incorporated into the network either as an additive or as a choice [14, 15], with the latter method adding a choice to the multiplicative noise [16]. In conclusion, using random weights is a hot research direction in adversarial defense or other tasks. However, among the existing task-heavy, one part is only applied to old or simple machine learning methods, such as SVM and LeNet, and the other part only uses random weights as a tool, instead of investigating the effect of random weights in deep neural networks on the forward and backward processes of the network [29, 30, 31]. In addition to this, there is also work in differential privacy that uses random weights[34]. Unfortunately, the design of random weights also does not take into account the optimization process with the network. While in the works above, the parameters of the random numbers, such as mean and variance, are typically pre-defined in these algorithms, Dredze et al. [35] have demonstrated that the parameters of the random sampling process can also be trained. Building on this insight, Kingma et al. [36] proposed a re-sampling-based method that enables the incorporation of trainable random weights into the neural network training process. Adversarial training is the most common method used by adversarial defense algorithms to achieve a robust model, and the early-stop algorithm has been proposed by Rice et al. [37] to prevent overfitting in this process. Adversarial training is widely employed by various adversarial defense algorithms [7, 13, 14, 15, 16]. In contrast to previous work, the focus of the present study is on training and constraining the parameters of the random sampling process, allowing for better adaptation of the process to the network and identifying an appropriate trade-off between accuracy under clean examples and adversarial attacks.

# 4 Experiments

## 4.1 Experiment Setup

We evaluate CTRW on CIFAR [38] and Imagenet [39]. We also include various networks for evaluation, including ResNet18, ResNet50 [17], and WRN34 [18].

**CIFAR** In the CIFAR-10 and CIFAR-100 datasets, there are 10 and 100 classes respectively. Each CIFAR dataset contains $5.0 \times 10^4$ training examples and $1.0 \times 10^4$ test examples, with all images scaled to $32 \times 32$ pixels in 3 channels. We implemented CTRW on both ResNet18 [17] and WRN34 [18] for CIFAR. In the experiment, the dataset is split into batches of size 128, and the weight decay is set to $5.0 \times 10^{-4}$. We use an SGD optimizer with a momentum of 0.9. The learning rate is initially set to 0.1 and reduced by a multi-step scheduler. The network is trained for 200 epochs, with the learning rate reduced by a factor of 10 at epochs 100 and 150. For adversarial training, we set $\epsilon_d$ to $\frac{8}{255}$ and the step length $\eta$ to $\frac{2}{255}$ for a 10-step PGD [9]. The bounds in CTRW are set to $\alpha = 1 \times 10^{-3}$

Table 3: Comparison with SOTA methods.

| Methods | CIFAR-10 | | Imagenet | |
|---|---|---|---|---|
| | PGD$^{20}$ | AA | PGD$^{10}$ | PGD$^{50}$ |
| RP-Ensemble [42] | 60.30 | - | - | - |
| RP-Regularizer [42] | 52.39 | - | - | - |
| RNA [14] | 63.34 | 67.88 | - | 54.61 |
| Additive Noise [15] | 62.36 | 58.47 | - | - |
| Overfit [37] | 55.06 | 52.24 | 39.85 | 39.19 |
| DWQ [43] | 52.18 | 49.70 | 42.88 | 42.72 |
| EnsTrans [44] | 67.00 | - | - | - |
| RobustWRN [45] | 59.13 | 52.48 | 31.14 | - |
| CTRW(ours) | **71.34** | **74.35** | **59.29** | **60.16** |

Table 4: Position Study.

| position in one layer | Natural | PGD$^{20}$ |
|---|---|---|
| in-channels | **83.69** | **69.27** |
| out-channels | 78.06 | 59.14 |
| both in- and out-channels | 72.69 | 49.52 |
| filters | 68.11 | 55.34 |

Table 5: Randomness of results on CIFAR-10.

| Baseline | Natural | PGD$^{20}$ |
|---|---|---|
| ResNet18 | $83.69 \pm 0.20$ | $69.27 \pm 0.28$ |
| WRN34 | $85.87 \pm 0.18$ | $70.87 \pm 0.31$ |

and $\beta = 1 \times 10^{-2}$. Thus, the lower bound of the sum of $\sigma$ can be derived as shown in Eq. 14, and in the experiment, it is estimated as $1 \times 10^3$. The model is implemented using PyTorch [40] and trained and evaluated on a single NVIDIA GeForce RTX 4090 GPU.

**Imagenet** The Imagenet dataset consists of $1.2 \times 10^6$ training examples and $5.0 \times 10^4$ test examples from 1000 different classes. The images in the dataset are 3-channel images with a size of $224 \times 224$ pixels. CTRW is implemented on ResNet50 [17]. In the experiment, Imagenet is split into batches containing 1024 examples. The weight decay and optimizer settings are the same as those in the CIFAR dataset, but the learning rate is set to 0.02 initially and managed by a cosine annealing scheduler. During adversarial training, the network is trained for 90 epochs, using a 2-step PGD [9] to generate adversarial examples. For the PGD attack, we set $\epsilon_d$ to $\frac{4}{255}$ and the step length $\eta$ to $\frac{1}{255}$. The settings of CTRW are the same as those in the CIFAR dataset. The model is implemented using PyTorch [40] and trained and evaluated on eight NVIDIA A100 Tensor Core GPUs.

**Adversarial Attacks** For CIFAR, we choose multiple adversarial attack algorithms, including black-box and white-box attacks from TorchAttacks [41]. The attack methods include PGD [9], FGSM [8], MIFGSM [20], Deep Fool [27], CW12 [28], and AA [10]. For PGD, we set the number of iterations to 20, and the other parameters are the same as in adversarial training. In FGSM, MIFGSM, and AA, $\epsilon_d$ is set to $\frac{8}{255}$, and in MIFGSM, the step length is $\frac{2}{255}$ with a decay of 0.1 over 5 iterations. The Deep Fool attack consists of 50 steps with an overshoot of 0.02. The CW12 attack is performed for 1000 steps with $c = 1.0 \times 10^{-4}$, a learning rate of 0.01, and $\kappa = 0$. For Imagenet, we evaluate a 10-step and a 50-step PGD attack with $\epsilon_d$ set to $\frac{4}{255}$

## 4.2 Results and Discussion

**Results on CIFAR-10/100** We demonstrate the effectiveness of CTRW against various attacks and compare it with other variants, as shown in Table 1 and Table 2. To assess the effectiveness of trainable $\mu$ and $\sigma$, as well as the constraints on $\sigma$, we added two additional sets of settings. One of them, labeled **without constraint**, represents the absence of any constraint on $\sigma$ during training. In this setting, we observe that for most pixels, $\sigma \to 0^+$, meaning that random weights converge to a fixed value. On the other hand, **fixed std** refers to fixing $\sigma$ at the lower bound without training, intending to maximize the probability that the network has low gradient similarity after each sample. On ResNet18, incorporating CTRW improves the accuracy of the network under PGD attack by $17.32\%$ on CIFAR-10 compared to the baseline, and by $13.3\%$ on CIFAR-100. Under Auto Attack, adding CTRW results in $25.87\%$ higher accuracy than the baseline on CIFAR-10 and $21.10\%$ higher on CIFAR-100. On WRN34, the improvement of CTRW under PGD is $16.28\%$ on CIFAR-10 and $12.64\%$ on CIFAR-100. Under AA, WRN34 with CTRW is $21.99\%$ more accurate than the baseline on CIFAR-10 and $22.11\%$ more accurate on CIFAR-100. To compare the case without bounds, we plotted the convergence of $\sigma$ as shown in Figure 3a. The number of models is divided into two groups: group 1 is number 1 to 4, which represents the models **without constrain**, while group 2 is number 5 to 8, which represents the models using CTRW. Specifically, within each group, the four models represent, in order, ResNet18 and WRN34 trained on CIFAR-10, as well as ResNet18 and WRN34 trained on CIFAR-100. It can be observed that the bounds cause $\sigma$ to converge to a smaller region with a higher value. These bounds prevent unbounded $\sigma$ values from degrading the random weights to fixed weights, resulting in better defense effectiveness of the random weights. When comparing CTRW with the **fixed std** setting, it can be seen that random weights have a greater advantage when $\sigma$ is trained. This is mainly attributed to the higher adaptability of trainable $\sigma$.

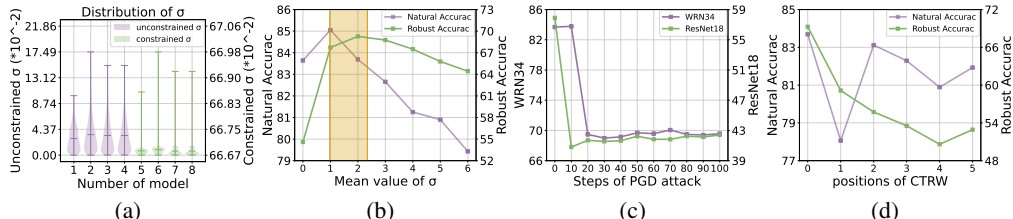

Figure 3: Evaluation under different settings: (a) The distribution of the final convergence of the standard deviation $\sigma$; (b) Comparison of accuracy over different $\sum \sigma$; (c) Robust Accuracy under different PGD strength; (d) Comparison of CTRW in different positions of one convolution layer.

**Effectiveness of Bound on Robust Accuracy**     According to the experimental setup on CIFAR, the lower bound is set to 1000. Based on the analysis in Section 2.2, using the network weights, we can approximate the upper bound as $B \approx 2400$. To verify the feasibility of the proposed bounds in this work, we set a range of values for $\sigma$, ranging from 0 to 6 times the image size. Among these values, 0 times the image size was set to 1 to prevent the degradation of random weights into fixed weights. The results are shown in Figure 3b, with the bound highlighted in orange. An interesting observation is that the average value of an optimal $\sum \sigma$ falls within the bound at 2048. It can be seen that once $\sum \sigma$ exceeds the lower bound, the robust accuracy experiences a significant drop compared to the natural accuracy. Moving from $\sum \sigma = 1024$ to 1, the natural accuracy drops from $85.05\%$ to $83.64\%$, while the robust accuracy experiences a significant drop from $67.72\%$ to $54.59\%$. Conversely, once $\sum \sigma$ exceeds the upper bound, both the robust accuracy and the natural accuracy decrease simultaneously. Beyond the upper bound, the gap between robust accuracy and natural accuracy is always around $14\%$. These experimental observations demonstrate that the bounds defined in Equation 14 effectively help $\sum \sigma$ take reasonable values for both two factors influencing robust accuracy.

**Comparison with Other Randomization Methods on CIFAR and Imagenet**     To demonstrate the superiority of CTRW, several other methods utilizing full random sampling were used for comparison. RP-Ensemble [13] trains different networks on multiple random mappings of the original input, which are then combined into an ensemble. RP-Regularizer [13] adds a regularization term to the original input to confuse adversarial attacks. The experiments on RP-Ensemble and RP-Regularizer were replicated by Carbone et al. [42], producing identical results. RNA [14] proposes a method of randomly selecting the normalization layer to map the random sampling process onto the network structure. Additive Noise [15] introduces an additive to the input data directly to confuse adversarial attack algorithms. DWQ [43] integrates random accuracy training and inference, reducing undesirable adversarial transferability through randomness to enhance model accuracy under adversarial attacks. EnsTrans [44] applies randomization to the input de-noising layer and output validation layer to resist internal attacks. RobustWRN [45] proposes a WRN framework for DNNs that is more suitable for adversarial defense. To compare with these models, we used state-of-the-art adversarial training strategies [37] for training. All models are compared on CIFAR-10 with WRN34. The results of the comparison are shown in Table 3. It can be observed that CTRW outperforms other adversarial defense algorithms under all attacks.

### 4.3   Ablation Study

**Effectiveness of Lower Boundaries**     In lemma1, $\alpha$ is specified to be less than 0. To verify the validity of this constraint, we count the number of samples that satisfy $cos(W^r[1], W^r[2]) \leqslant \epsilon_r$ when joining the constraint versus not joining the constraint, respectively. Samples that satisfy the condition $cos(W^r[1], W^r[2]) \leqslant \epsilon_r$ are called positive samples (PS), and the results are shown in Table 6. Based on the statistical results, it can be seen that without constraints, the number of samples that satisfy the condition is 0, that is, $P(cos(W^r[1], W^r[2]) \leqslant \epsilon_r) = 0$. While with constraints, the number of samples that satisfy the condition among $1e5$ samples are all $4e4$ or more. That is, with the addition of the constraint, more samples have cosine similarity satisfying the condition of less than $\epsilon$. It can also be seen that $0.5 > P(cos(W^r[1], W^r[2]) \leqslant \epsilon_r) > F(\alpha)$. That is, the assumption that $\alpha < 0$ is also satisfied. In addition to this, it can be seen that the constraint in lemma1 effectively reduces the cosine similarity based on the minimization of the cosine similarity.

**Robust Accuracy under Stronger PGD Attacks**     To verify the effectiveness of CTRW under PGD attacks of varying strength, we adjusted the number of iteration steps in PGD attacks and tested the model against PGD from 0 steps (i.e., the clear situation) up to 100 steps, with 10 iterations added

| Dataset | Constrain | Number of PS | Ratio of PS | $\min(cos(W^r[1], W^r[2]))$ | Acc |
|---------|-----------|--------------|-------------|------------------------------|-----|
| CIFAR10 | with (ours) | 4558 | 0.4558 | -0.0557 | 69.48 |
| CIFAR10 | without | 0 | 0.0000 | 0.7835 | 54.59 |
| CIFAR100 | with (ours) | 4124 | 0.4124 | -0.0669 | 42.01 |
| CIFAR100 | without | 0 | 0.0000 | 0.7361 | 30.05 |

Table 6: Effect of adding or not adding constraints on $P(cos(W^r[1], W^r[2]) \leqslant \epsilon_r) > F(\alpha)$

each time. We used CIFAR-10 as the dataset and ResNet18 as the baseline in this ablation study. The results are shown in Figure 3c. According to the curves in the figure, it can be observed that CTRW remains stable at $69\% \pm 1\%$ for all PGD attacks with more than 10 steps.

**Position of CTRW** Considering that the upper and lower bounds of $\sigma$ may differ on different layers, we added random weights to different layers and compared the results. We loaded CTRW at multiple positions in ResNet18 and also added random weights to different positions in the first-layer convolution of ResNet18. It is important to note that all experiments in this section were conducted on CIFAR-10. Specifically, we changed the position of CTRW to the front of each residual layer and the Fully Connected (FC) layer in ResNet18 [17]. In Figure 3d, the numbers in the "position" column denote the number of the residual layer adjacent to CTRW, where $0$ represents the first convolutional layer and $5$ corresponds to the FC layer. We observed that loading CTRW earlier in the network provides more assistance in terms of robust accuracy. Thus, the best position for CTRW in the entire network is at the first convolutional layer. Furthermore, the position of CTRW around the convolutional layer is still a question to be considered. We explored various ways of adding the random weights. Apart from adding the weights before the in-channels of the convolutional layer, we also tried the out-channels and both in- and out-channels. Additionally, we placed the random weights into the convolutional layer on the filter. The results are presented in Table 4. We tested these methods under PGD-20 and clean examples without altering other settings. We found that placing the random weights in the in-channel yields the best performance. It is observed that more random weights do not lead to better performance. Therefore, to ensure the effectiveness of CTRW, the random weights should only influence the input feature map of the selected convolutional layer.

**Randomness of Evaluation** According to the analysis in Section 2.2, the inference results may vary each time. Therefore, to verify the accidental results, we repeated the evaluation on CIFAR-10 multiple times. We recorded the means and standard deviations of the natural and PGD20 evaluations, which are shown in Table 5. It can be observed that each network exhibits low standard deviations, indicating that the results of this work are not accidental.

**Evaluation of fixed $\sigma$** To further explore the significance of the values of $\sigma$, we fix the values of $\sigma$ to the upper bound, the lower bound, and the middle of the upper and lower bounds, respectively, and compare them with CTRW. The results are shown in Table 7.

| Value of $\sigma$ | $PGD^{20}$ |
|-------------------|------------|
| Fixed to lower bound | 67.75 |
| Fixed to upper bound | 66.06 |
| Fixed to mid value | 67.50 |
| CTRW (ours) | 69.48 |

Table 7: Robust accuracy for different values of $\sigma$.

## 5 Conclusion

In this paper, we investigate the role of trainable and restricted random weights in defending against adversarial attacks. We argue that randomness can enhance the variability of gradients between inference processes, but can also reduce the overall performance of the network. We designed CTRW and analyzed the relationship between the bounds of $\sigma$ and its robust accuracy, and proved the validity of the bounds. CTRW is found to outperform other randomized methods. The results of this work provide valuable insights for developing strategies to exploit randomization in adversarial defenses.

## Acknowledgments

This work was supported in part by the Australian Research Council under Projects DP210101859 and FT230100549.

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
