# Supplementary Material for Adversarial Robustness through Random Sampling under Constraints

**Yanxiang Ma, Minjing Dong, Chang Xu**[*]
School of Computer Science, University of Sydney
{yama9404, mdon0736}@uni.sydney.edu.au
c.xu@sydney.edu.au

## I  Proof of Lower Bound

**Lemma 1.** *Considering a fixed $W^r[1]$ with size of $m$, the constraint $cos(W^r[1], W^r[2]) \leqslant \epsilon_r$ where $\epsilon_r = o(1)$ holds with the probability at least $F(\alpha)$, where $F$ denotes the cumulative function of a standard Gaussian distribution. $P(cos(W^r[1], W^r[2]) \leqslant \epsilon_r) > F(\alpha)$, if*

$$\sum_{i=1}^{m} \sigma_i \geqslant \frac{\epsilon_r' - \sum_{i=1}^{m} \mu_i}{\alpha \sum_{i=1}^{m} |W^r[1]_i|} \geqslant 0, \tag{1}$$

*where for any variation $a$, $a_i$ denotes the $i$th element of $a$, $m$ is the size of $W^r$, $\epsilon_r' = o(1)$ and $\epsilon_r' = \epsilon^r \times \|W^r[1]\|_p \|W^r[2]\|_p$.*

*Proof.* First, we define the cosine similarity of gradients of $h[1]$ and $h[2]$ over $x$ as,

$$cos(\nabla(x, W^r[1]), \nabla(x, W^r[2])). \tag{2}$$

According to the chain rule of derivation, the cosine similarity can be transformed into,

$$cos(\nabla^{r+1,n}(x, W^r[1])\nabla^{r,r+1}(x, W^r[1])\nabla^{1,r}(x, W^r[1]),$$
$$\nabla^{r+1,n}(x, W^r[2])\nabla^{r,r+1}(x, W^r[2])\nabla^{1,r}(x, W^r[2])), \tag{3}$$

where $\nabla^{i,j}(x, W^r)$ denotes the gradient from layer $j$ to layer $i$ over $x$. The angle $\langle \nabla(x, W^r[1]), \nabla(x, W^r[2]) \rangle$ can be divided into,

$$\langle \nabla(x, W^r[1]), \nabla(x, W^r[2]) \rangle = \langle \nabla^{r+1,n}(x^{r+1}, W^r[1]), \nabla^{r+1,n}(x^{r+1}, W^r[2]) \rangle$$
$$+ \langle \nabla^{r,r+1}(x^r, W^r[1]), \nabla^{r,r+1}(x^r, W^r[2]) \rangle \tag{4}$$
$$+ \langle \nabla^{1,r}(x, W^r[1]), \nabla^{1,r}(x, W^r[2]) \rangle$$

Since the weights in layer 1 to layer $r$ are all the same, $\langle \nabla^{1,r}(x, W^r[1]), \nabla^{1,r}(x, W^r[2]) \rangle$ is related only to $x^r$, where $x^i$ is the input feature map of layer $i$, and $x$ is $x^1$. We can use $\langle \nabla^{r,n}(x^r, W^r[1]), \nabla^{r,n}(x^r, W^r[2]) \rangle$ to represent $\langle \nabla^{1,n}(x, W^r[1]), \nabla^{1,n}(x, W^r[2]) \rangle$. Thus Eq. 4 can be simplified as,

$$\langle \nabla(x^r, W^r[1]), \nabla(x^r, W^r[2]) \rangle = \langle \nabla^{r,r+1}(x^r, W^r[1]), \nabla^{r,r+1}(x^r, W^r[2]) \rangle$$
$$+ \langle \nabla^{r+1,n}(x^{r+1}, W^r[1]), \nabla^{r+1,n}(x^{r+1}, W^r[2]) \rangle \tag{5}$$

---

[*]Corresponding author.

37th Conference on Neural Information Processing Systems (NeurIPS 2023).

Let $cos(\nabla^{1,n}(x, W^r[1]), \nabla^{1,n}(x, W^r[2])) \leqslant \epsilon_R$, there must exists am $\epsilon_r$ that satisfies $cos^{r,n}(\nabla(x^r, W^r[1]), \nabla^{r,n}(x^r, W^r[2])) \leqslant \epsilon_r$. As written in Eq. 5, the cosine similarity can be expanded as,

$$cos(\nabla^{r,n}(x^r, W^r[1]), \nabla^{r,n}(x^r, W^r[2])) = cos(\nabla^{r,r+1}(x, W^r[1]), \nabla^{r,r+1}(x, W^r[2]) \\ + \nabla^{r+1,n}(x^{r+1}, W^r[1]), \nabla^{r+1,n}(x^{r+1}, W^r[2])) \quad (6)$$

Using the sum-angle formula for the cosine function, 6 can be expanded as,

$$cos(\nabla(x, W^r[1]), \nabla(x, W^r[2])) = cos(\nabla^{r,r+1}[1], \nabla^{r,r+1}[2])cos(\nabla^{r+1,n}[1], \nabla^{r+1,n}[2]) \\ - sin(\nabla^{r,r+1}[1], \nabla^{r,r+1}[2])sin(\nabla^{r+1,n}[1], \nabla^{r+1,n}[2]), \quad (7)$$

as shown in the main text, where $\nabla^{a,b}[i]$ denotes $\nabla^{a,b}(x^a, Wr[i])$. We can obtain that $\langle \nabla^{r+1,n}[1], \nabla^{r+1,n}[2] \rangle$ is related only to the angle of the outputs $\langle y[1], y[2] \rangle$. Assume that $y_1 - y_2 \leqslant \epsilon_y$ where $\epsilon_y = o(1)$, *i.e.* the angle $\langle y[1], y[2] \rangle \leqslant \epsilon^y_{ang}$ where $\epsilon^y_{ang} = o(1)$. Thus, $sin(\nabla^{r+1,n}[1], \nabla^{r+1,n}[2]) = o(1)$, and $cos(\nabla^{r+1,n}[1], \nabla^{r+1,n}[2])$ is close to 1 . Now that Eq.7 can be simplified by approximation as,

$$cos(\nabla(x, W^r[1]), \nabla(x, W^r[2])) \approx cos(\nabla^{r,r+1}[1], \nabla^{r,r+1}[2]). \quad (8)$$

So, we can approximate the bound of $cos(\nabla^{r,r+1}[1], \nabla^{r,r+1}[2])$ as $cos(\nabla^{r,r+1}[1], \nabla^{r,r+1}[2]) \leqslant \epsilon_r$. Since $\nabla^{r,r+1} = W^r$, we can find that $cos(\nabla^{r,r+1}[1], \nabla^{r,r+1}[2]) = cos(W^r[1], W^r[2])$. As is defined in the main text, $W^r = \mu + \delta \odot \sigma$. Once the attack is accomplished, $W^r[1]$ is fixed, so that the cosine similarity can be expanded as,

$$cos(\nabla^{r,r+1}[1], \nabla^{r,r+1}[2]) = \frac{W^r[1] \cdot (\mu + \delta[2] \odot \sigma)}{\|W^r[1]\|_p \|W^r[2]\|_p} \leqslant \epsilon_r \quad (9)$$

where $\epsilon_r = o(1)$, $\delta[i]$ is the $\delta$ in $W^r[i]$, and $m$ is the size of $W^r$. Thus, to make 9 hold with a probability of $F(\alpha)$, we must let the upper bound of $\sigma$ bigger than $\alpha$, *i.e.*

$$\alpha \leqslant \frac{\epsilon'_r - \sum_{i=1}^m \mu_i}{\sum_{i=1}^m \sigma_i \sum_{i=1}^m |W^r[1]_i|}, \quad (10)$$

where $\epsilon'_r$ is $\epsilon_r \times \|W^r[1]\|_p \|W^r[2]\|_p = o(1)$. Since $\epsilon_r = o(1)$, we have $\epsilon'_r < \mu$. Since $\sigma_i$ and $|W^r_i|$ are all positive, we have that $\alpha$ is negative, so Eq.9 can be transferred as the lower bound of $\sigma$, where,

$$\sum_{i=1}^m \sigma_i \geqslant \frac{\epsilon'_r - \sum_{i=1}^m \mu_i}{\alpha \sum_{i=1}^m |W^r[1]_i|}. \quad (11)$$

Thus, the lower bound of $\sigma$ is set as Eq.11 $\qquad\qquad\qquad\qquad\qquad\qquad\qquad\qquad$ □

## II  Proof of Upper Bound

Only the highest confidence term in the output of the neural network affects the classification result. The change in the predicted outcome is defined according to the difference in the confidence of the ground-truth term in the two outputs. To make the change in the prediction result smaller than the minimal term $\epsilon_y$. The difference between two outputs $\Delta y_k$ can be defined as,

$$\Delta y_k = y[1] - y[2] \leqslant \epsilon_y. \quad (12)$$

Using the function of the network to represent $y$, we have,

$$\Delta y_k = \mathcal{H}^{r+1,n}(W^r[1] \odot \mathcal{H}^{1,r}(x)) - \mathcal{H}^{r+1,n}(W^r[2] \odot \mathcal{H}^{1,r}(x)) \leqslant \epsilon_y, \tag{13}$$

where $\mathcal{H}^{i,j}$ denotes the function of network from layer $i$ to layer $j$. For linear part in $\mathcal{H}$, $\mathcal{H}^{i,j}(W^r \odot x) = \mathcal{H}^{i,j}(W^r) \odot \mathcal{H}^{i,j}(x)$. For the nonlinear part, which is always defined as $ReLU$, we have,

$$ReLU(a \odot b) \begin{cases} > ReLU(a) \odot ReLU(b), & \text{if } a < 0 \text{ and } b < 0, \\ = ReLU(a) \odot ReLU(b), & \text{if } a > 0 \text{ or } b > 0. \end{cases} \tag{14}$$

Thus, we can approximate that in DNNs, $\mathcal{H}^{i,j}(W^r \odot x) = \mathcal{H}^{i,j}(W^r) \odot \mathcal{H}^{i,j}(x)$. Then we can have Eq.13 expanded as,

$$\Delta y_k = \mathcal{H}^{r+1,n}(W^r[1]) \odot \mathcal{H}^{r+1,n}(\mathcal{H}^{1,r}(x)) - \mathcal{H}^{r+1,n}(W^r[2]) \odot \mathcal{H}^{r+1,n}(\mathcal{H}^{1,r}(x)) \leqslant \epsilon_y. \tag{15}$$

Since each standard Gaussian distribution is *i.i.d.*, the sum of the two sampled results is also a Gaussian distribution, whose mean value and variation are added. This property of the Gaussian distribution is known as convergence. Using the convergence of the Gaussian distribution, it can be defined that $\mathcal{H}^{r+1,n}(W^r) \sim N(\mu', \tilde{\sigma})$. The calculation process of $\mu'$ and $\tilde{\sigma}$ is introduced in the main text. Thus, except for the Fully Connection layer (FC layer), $\mathcal{H}^{r+1,n-1}(W^r) \sim N(\mu', \tilde{\sigma})$ can be represented as $\mu' + \delta \odot \tilde{\sigma}$. Let $x^{n-1}$ be the input feature map of the FC layer in the network with no random weight, and $W^n$ be the weight in the FC layer, and the difference can be expressed as,

$$\Delta y_k = (\tilde{\sigma} \odot (\delta[1] - \delta[2])) \odot x^{n-1}) \cdot W^n \leqslant \epsilon_y. \tag{16}$$

It can be obtained that $\delta[1] - \delta[2] \sim N(0, 2)$, whose standard deviation is $\sqrt{2}$. SO that if $P(\delta[1] - \delta[2] < \beta) = F(\frac{\sqrt{2}\beta}{2})$. To make $P(\Delta y_k < \epsilon_y) \geqslant F(\frac{\sqrt{2}\beta}{2})$, we have,

$$P(\tilde{\sigma} \odot (\delta[1] - \delta[2])) \odot x^{n-1}) \cdot W^n \leqslant \epsilon_y) \geqslant F(\frac{\sqrt{2}\beta}{2}). \tag{17}$$

Rearranging the Eq.17, we can get,

$$P((\delta[1] - \delta[2]) \leqslant \frac{\epsilon_y}{(\tilde{\sigma} \odot x^{n-1}) \cdot W^n} \geqslant F(\frac{\sqrt{2}\beta}{2}). \tag{18}$$

Let $\sigma'_i = \tilde{\sigma}_i \times W^n_i$, and according to the definition of the cumulative function of the Gaussian distribution, we can obtain that,

$$\beta \leqslant \frac{\epsilon_y}{\sum_{i=1}^m \sigma'_i (x^{n-1} \cdot W^n)} \tag{19}$$

where $m$ is the size of $W^n$. Since $x^{n-1} \cdot W^n$ is the highest element in the output of the original network, we can consider that $x^{n-1} \cdot W^n > 0$. Thus, each term on the right-hand side of the formula 19 is positive, and we can obtain the upper bound as,

$$\sum_{i=1}^m \sigma'_i \leqslant \frac{\epsilon_y}{x^{n-1} \cdot W^n \beta}, \tag{20}$$

## III    Comparison with Random Defence Methods by ResNet for CIFAR

Besides those SOTA methods in the main text, we also compared CTRW with PNI[1] and Adv-BNN[2] and Learn2Perturb[3] These methods are implemented using the baseline of ResNet for

| Model | baseline | | CTRW (ours) | | PNI[1] | | Adv-BNN[3] | | Learn2Perturb[3] | |
|---|---|---|---|---|---|---|---|---|---|---|
| | Natural | PGD | Natural | PGD | Natural | PGD | Natural | PGD | Natural | PGD |
| ResNet20[4] | 75.34 | 48.62 | 79.62 | **66.28** | **84.90** | 45.90 | 65.76 | 44.95 | 83.62 | 51.13 |
| ResNet32[4] | 77.66 | 48.77 | 81.21 | **67.67** | **85.90** | 43.50 | 62.95 | 54.62 | 84.19 | 54.62 |
| ResNet44[4] | 80.35 | 49.60 | 82.39 | **68.36** | 84.70 | 48.50 | 76.87 | 54.62 | **85.61** | 54.62 |
| ResNet56[4] | 80.91 | 50.27 | 82.18 | **68.77** | **86.80** | 46.30 | 77.20 | 54.62 | 84.82 | 54.62 |

Table 1: Comparasion on CIDAR-10 with other methods using ResNet for CIFAR[4]

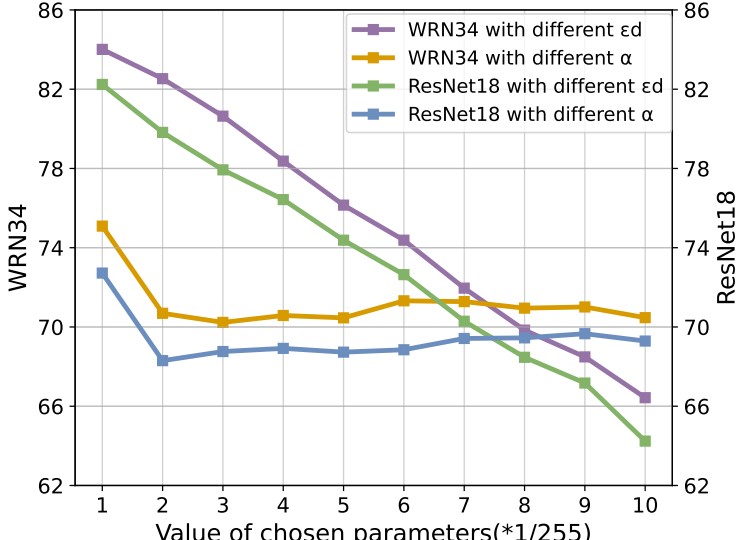

Figure 1: Evaluation under different $\epsilon_d$ and $\alpha$

CIFAR[4] such as ResNet20, ResNet32, ResNet44, and ResNet56. This series of ResNet has a smaller feature map in the middle than the original ResNet such as ResNet18, and performs better on clean samples of CIFAR-10. Therefore, to exclude the interference, this work implements CTRW on the ResNet-20 series and compares it with existing work. The results are shown in Table 1. CIFAR-10 is used as the dataset and all the settings are the same as in the main text. As can be seen, our method has the highest robust accuracy under PGD attack, and the natural accuracy is at the middle level compared to other methods.

## IV Evaluation under Different Attack Strength

As mentioned in the main text, we have tested our model under different steps of PGD attacks[5]. however, there are still two other parameters that influence the strength of PGD attack. One is $\epsilon_d$ and another is the step length $\alpha$. The higher these two parameters are, the more strong the attack is. To test the defense capability under a different situation, we evaluated CTRW under PGD with $\epsilon_d$ from 1 to 10 and with $\alpha$ from 1 to 10. The results are shown in Figure 1

## V EOT Evaluation

The sampling process of random weights was set to sample 1 times per iteration step in adversarial training and once per batch of data in defense. For the attack algorithm, multiple iterations in a round of attack are randomly sampled only once. To verify the performance of the model under EOT, we re-trained and evaluated the model with CTRW added on CIFAR-10 using ResNet18 as the baseline using the EOT method. We set the PGD to randomly sample once per iteration and kept the other experimental settings as in the main text. Compared to the baseline, the accuracy of our model improved by 6.8% under EOT and decreased by 13.4% compared to the model trained under non-EOT. Since the random sampling process has minimized the correlation between the gradient at the time of attack and the gradient at the time of inference as much as possible. So resampling

in each attack iteration step may lead to a shift in the gradient, instead the correlation between the gradient at attack time and at inference time increases. This explains one reason for the performance degradation after increasing the sampling frequency.

## VI  Evaluation of Black-Box Attack

We include more evaluation of black-box attacks, such as Square and Pixle. We conducted an evaluation of ResNet-18 on CIFAR-10. The robust accuracy is shown in Table 2. It is clear from the table that CTRW has an equally strong defense against black-box attacks.

| Model | Square | Pixle |
|---|---|---|
| baseline | 54.68 | 8.10 |
| CTRW(ours) | 77.73 | 72.14 |

Table 2: Comparison under black box attack

## VII  Evaluation of model without adversarial training

It is interesting to see the performance without adversarial training. We evaluated with ResNet-18 on CIFAR-10. All the algorithms are trained with natural training. The results are shown in Table 3. According to the table, it can be seen that confrontational training is very important for CTRW. It can also be seen that networks that incorporate the designed random weights are inherently robust, although this is not significant.

| Model | Natural | $PGD^{20}$ |
|---|---|---|
| baseline | 84.17 | 0.00 |
| CTRW(ours) | 84.61 | 3.44 |

Table 3: Results of models without adversarial training

## VIII  Evaluation on Vision Transformer

Our proposed algorithm can be simply deployed to other neural networks, such as Vision Transformer (ViTs)[6]. For illustration, we deploy CTRW on Vision Transformer-Small (ViT-S) and evaluate the performance on CIFAR-10. The results are shown in Table 4. The experimental results demonstrate the better performance of CTRW even on deeper networks.

| Model | cw20 | $PGD^{20}$ |
|---|---|---|
| baseline | 34.62 | 33.49 |
| CTRW(ours) | 45.21 | 45.68 |

Table 4: Results of ViT[6] based adversarial robust models