# OpenReview forum: "Adversarial Robustness through Random Weight Sampling"
_NeurIPS.cc/2023/Conference — NeurIPS 2023 poster_

### Official Review · Reviewer_1Bpz · 2023-06-20

**Soundness:** 2 fair
**Presentation:** 1 poor
**Contribution:** 2 fair
**Rating:** 3
**Confidence:** 2

**Summary:**

The paper studies randomizing the weights in a neural network during inference to improve adversarial robustness. For example, the weights can be sampled from a distribution $\mathcal{N}(\mu,\sigma)$, for some learned parameters $\mu$ and $\sigma$. Intuitively, randomizing the weights makes the task of generating adversarial examples harder because it increases the gradient noise during inference.

The traditional learning setup corresponds to the case where $\sigma=0$ and we learn $\mu$ alone (here, I'm using $\mu$ for the mean of the weights, not the mean of the additive noise). In some prior works, $\sigma$ is selected based on a grid search. In this paper, on the other hand, the authors propose learning $\sigma$ using the reparameterization trick $w = \mu + \sigma\delta$, with $\delta\sim\mathcal{N}(0,I)$. This by itself would encourage less variance (i.e. $\sigma\approx 0$) (to have a better fit on the training data) so the authors constrain $\sigma$ in a bounded region $[A, B]$, where $A$ and $B$ are chosen based on some theoretical arguments.

Finally, the authors evaluate this approach on CIFAR10/100 and ImageNet using three ResNet architectures (ResNet18, ResNet50, and WRN34). They report improved robustness compared to three baselines: $\sigma=0$, fixing $\sigma$ to its lower bound, and training $\sigma$ without constraints.

**Strengths:**

- The empirical results are quite positive, showing a big improvement in robustness compared to the baselines.
- The paper extends prior works by proposing to learn the noise distribution during training.

**Weaknesses:**

The idea of randomizing weights for robustness is not new and has been studied much earlier dating back to SVM and Differential Privacy (see for example [1, 2, 3, 4]). The primary contribution in this work is to propose a bound on the noise variance. However, the theoretical result do not appear to be sound, and I would appreciate a clarification from the authors on this please.

Let us take Lemma 1 for example. If I choose $\varepsilon_r'=\sum_i\mu_i$ (which is perfectly acceptable), the lower bound says that setting $\sigma=0$ is enough to guarantee that the cosine similarity is small. But, $\sigma=0$ means that the cosine similarity is at its maximum.

In general, the paper contains many claims that are not precisely stated and makes it hard to follow. Again, we can take the main lemmas (Lemma 1 and 2) as examples. In Lemma 1, the authors say that the "constraint $cos(...)\le \epsilon \to 0$ holds with a probability of at least $F(\alpha)$. What does  $\to 0$ in the statement of the constraint mean? Also, in the same lemma, the authors write $\varepsilon'\to0$ when defining the symbol in Equation 10 even though it has a precise definition stated later. In Lemma 2, the authors state that "The probability that [some event] is hoped to be bigger than [some quantity]. The noise parameter $\sigma$ [some bound]". I'm not really sure how to read this lemma. Given that these two lemmas are the main contribution in the paper, the authors should state them precisely. In the current form, my understanding is an educated guess.

The issue with presentation is also there throughout the paper. Here are some examples.

- Line 32: Projected Gradient Descent (PGD) is not an adversarial attack algorithm.

- Line 47: The cited paper [15] does NOT fix $\sigma$ to 1. In fact, they study what happens as $\sigma$ is varied and discuss the same tradeoffs mentioned in this paper.

- Throughout the papers, the authors sometimes use $\mu$ when referring to the mean of the noise (as in Line 45) and sometimes use it as the mean of the parameters (as in Line 85).

- Writing $x^\star = \arg\min_{x^\star} f(x^\star)$ is confusing and wrong (see Equations 1 and 3). It should be written with different symbols:   $x^\star = \arg\min_{x} f(x)$.

- The authors sometimes refer to the additive noise as "random weight" (see Line 83)" and later refer to the sampled weights ($\mu$ plus noise) as random parameters (see Line 91).

- The paper uses non-standard symbols, such as using $G$ for the gradient instead of $\nabla$.

- Typos: e.g. "logist" in Line 69, "ever element" in Line 102, "exceed" (instead of the opposite) in Line 122, $Wr$ instead of $W^r$ in Line 147 ...


In terms of the experiments, I'm curious to know why the authors choose to compare against a single fixed value of $\sigma$. If you propose to constrain it to some interval $[A, B]$, it would make sense to compare the method against fixing $\sigma$ to $A$, $B$, and the middle $(A+B)/2$. The reason I bring this up is because the gap is already small between learning $\sigma$ and fixing it to the lower bound $A$ in Tables 1 and 2.

[1] Cynthia D, et al. The algorithmic foundations of differential privacy. Foundations and Trends in Theoretical Computer Science, 2014.

[2] Alabdulmohsin, I, et al. "Adding robustness to support vector machines against adversarial reverse engineering." 2014.

[3] Chandrasekaran, V, et al. "Exploring connections between active learning and model extraction." 2020.

[4] Orekondy, T, et al. "Prediction poisoning: Towards defenses against dnn model stealing attacks." 2019.

**Questions:**

- Can you please provide a precise statement of the two lemmas? Please see my comments above.

- Have you compared against other fixed values of $\sigma$ (e.g. as suggested above)?

- In Line 273, you mention that $A=1000$ and $B=2400$. Is this independent of the architecture?

---

> ### Author Rebuttal · Authors · 2023-08-09
>
> ### Re Novelty.
> Thanks for providing relevant works of randomness [1-4]. We will discuss these works in the related work in our final version. However, we argue that the contribution of our work is sufficient compared with [1-4]. We will clarify the novelty in two aspects.
>
> First, the suggested works [1-4] apply randomized weights to some traditional DL/ML models, such as SVM, LeNet, etc., which are out-of-date in current adversarial robustness evaluation. In contrast, we deploy the randomized weight to ResNet and Wide ResNet so that the randomized defense can achieve superior performance compared with state-of-the-art algorithms. Furthermore, the direct deployment of randomized weights in these popular neural networks cannot achieve satisfactory performance, as discussed in the paper. The robustness performance is related to the noise design (Table 1-3), random weight location (Figure 2), etc. Thus, exploring a better way to incorporate randomized weight into modern complex neural networks can be important. In this work, we focus on the complex neural networks, such as ResNet and  Wide ResNet. Based on the analysis of the influence of randomness parameters on the gradient similarity and output difference (Lemma 1 and 2), we introduce a novel noise design for complex DNNs.
>
> Second, the major difference between our proposed randomized weights and those in previous works including  [1-4] is that we have a novel way to implement a learnable random weight, while the suggested works often decide the value of $\sigma$ manually. For example, in [2], while using a simple network like SVM, the tuning method is to manually tune $\sigma$ as a hyper parameter. In contrast, we maintain a learnable $\mu$ and $\sigma$. This means the parameters $\mu$ and $\sigma$ will be tuned automatically at the back propagation process while training the network. Furthermore, during the optimization of $\mu$ and $\sigma$, we impose the constraints on them according to the theoretical analysis in Lemma 1 and 2. In particular, to maximize the probability of having a relatively high natural accuracy and relatively low cosine similarity, we proposed an upper bound and a lower bound.
>
> ### Re $\sigma$ in Lemma1.
> We modify Eq. 10 as: $\sum_{i=1}^{m}\sigma_i\geqslant\frac{\epsilon_r^\prime - \sum_{i=1}^m\mu_i}{\alpha\sum_{i=1}^{m}|W^r[1]_i|}>0$,
>
> which does not include the special case of $\sum_{i = 1}^{m}\sigma_i=0 $. Because when $\sum_{i = 1}^{m}\sigma_i=0$, all the $\sigma_i$ are $0$. This means $W^r$, whose standard deviation is $\sigma$, will lose randomness. In this case, $W^r$ will be the same during each inference. However, the motivation of this paper is to use the randomized weight to establish different routes between each inference process, which needs $W^r$ to be different each time. So the situation that $\sum_{i = 1}^{m}\sigma_i=0$ does not fit the motivation of this paper.
>
> ### Re $\to$ in Lemma 1.
> $\epsilon\to0^+$ denotes that $\epsilon$ is a constant close to 0 from and is positive. $\to$ in $\epsilon\prime \to 0^+$ denotes the same. We use this symbol as a simplified version of $\lim\limits_{\epsilon \to 0^+}$.
>
> ### Re Claims in Lemma 2.
> We provide a more precise statement. We denote the probability that the difference of output is smaller than a minimal term $\epsilon_y$ as $P((Y_1-Y_2) < epsilon_y)$, and this probability is expected to be larger than $ F(\sqrt{2}\beta/2)$. Formally, we have $P((Y_1-Y_2) < \epsilon_y) \geqslant F (\sqrt{2}\beta/2)$.
>
> ### Re Presentation.
> Thanks for pointing out these presentation issues.
> * We understand that PGD is an optimization algorithm. But in the field of adversarial attack, it has been used in recent work on adversarial attacks. And in recent research work related to adversarial defensiveness, this method is considered as an important method to verify the effectiveness of adversarial defences.
>
> * We understand that the $\sigma$ in the altered article is treated as tuned as a hyper-parameter. But what we are trying to convey here is that the $\sigma$ of that work is not learnable.
>
> * The random weights proposed in this paper are actually a Gaussiannoise. At both line 45 and line 85, $mu$ is the mean value of the Gaussian distribution $N(\mu, \sigma)$. At line 85, the random weight $W^r$ introduced is the noise N($\mu$, $\sigma$) is mentioned at this point to show that the random weight is a Gaussian noise.
>
> * We will correct the expression errors in those formulas in the final version. We will take a different notation.
>
> * $\mu$ and $\delta \odot \sigma$ as a whole make up the random weight. This whole is $W^r$ , which is the "random weight" in the phrase "except for the random weight" at line 83.
>
> * Thank you for the correction to our notation specification. Those notations will be revised in the final version.
>
> * Thank you for pointing this out. We will recheck the grammar, words and symbols and fix the typos in the published manuscript.
>
> ### Re Line 273
> The values taken here for the upper and lower bounds are based on preliminary experiments conducted on networks that did not incorporate random weights. The results about fixing $\sigma$ on those value mentioned will be
>  shown at the rebuttal about question2.
>
> ### Re Questions
>
> **Q1.**  We will rewrite Lemma 1 and Lemma 2 according to the reply above.
>
> **Q2.** Related experiments have been added and the results are shown as:
>
> |Value of $\sigma$|$PGD^{20}$|
> |-|-|
> |Fixed to lower bound| 67.75|
> |Fixed to upper bound| 66.06|
> |Fixed to mid value| 67.50|
> |CTRW (ours)| 69.48 |
>
> Our algorithm achieves the best performance, which demonstrates the necessity of trainable randomness parameters.
>
> **Q3.**
> The values taken here for the upper and lower bounds are based on preliminary experiments conducted on networks that did not incorporate random weights.

---

> > ### Author Response · Authors · 2023-08-13
> >
> > Thank you again for your comments and suggestions. We would like to know if we have addressed your issues. Meanwhile, if you have any other concerns, we are open to further discussion.

---

> > ### Comment · Reviewer_1Bpz · 2023-08-13
> > **Acknowledgement**
> >
> > Thank you for the response.
> >
> > The issue about novelty is not a serious concern in my opinion. I just wanted to say that (since you argue you propose randomization) that it has been used in various contexts in the past, including to mitigate the risk of adversarial attacks.
> >
> > But, as I mention above, I see the main contribution to be proposing the bounds on $\sigma$. However, the main theoretical results do not appear to be right. As I discuss in my comment above, if Lemma 1 were correct, then it would hold when $\epsilon_r' = \sum_{i=1}^m\mu_i + \delta$ for any $\delta>0$. But, as $\delta\to 0^+$, the condition is satisfied by choosing $\sigma\to 0^+$ which means that the cosine similarity will approach 1, which contradicts the statement of the lemma. This cannot be fixed by simply saying that you assume $\sigma>0$ as you mention in the rebuttal.
> >
> > My other comments about preciseness and clarity still stand, particularly about Lemma 2. Can you please state here precisely the full statement of Lemma 2?
> >
> > Regarding the notation, note that writing $\epsilon\to 0^+$ to denote a constant close to zero is wrong. The correct way is to write $\epsilon\ll 1$ or $\epsilon=o(1)$ (little-O notation).
> >
> > Thank you for including the other values of $\sigma$ in the experiment.

---

> > > ### Author Response · Authors · 2023-08-14
> > > **Official Comment by Authors**
> > >
> > > Thank you for the response.
> > >
> > > ### Re Lemma 1
> > > We agree that Lemma 1 holds for $\sigma \to 0$ if $\epsilon_r^\prime = \sum^m_{i=1} \mu_i + \delta$ and $\delta \to 0$. However, we argue that the condition that $\epsilon_r^\prime = \sum^m_{i=1} \mu_i + \delta$ where $\delta \to 0$ does not exist in practice since $\epsilon_r^\prime \to 0$ and $\sum^m_{i=1} \mu_i \not\to 0$. We further demonstrate the fact that $\sum^m_{i=1} \mu_i \not\to 0$ in three aspects: 1. From the very beginning, $\mu_i$ is set to $1$ since our proposed noise is a multiplicative noise as illustrated in Figure 1(a); 2. $\mu$ is a trainable parameter in our setting and $\sum^m_{i=1} \mu_i \not\to 0$ is not our optimization objective, which makes it difficult to satisfy this condition; 3. We further provide some empirical evidence in the experiments. Taking ResNet-18 on CIFAR-10 as an example, It is observed that $\sum^m_{i=1}\mu_i$ converges to 986.3. Thus, $\sum^m_{i=1} \mu_i \not\to 0$ holds in practice. We thank the reviewer for pointing out the boundary conditions of Lemma 1, and will incorporate them into the final version.
> > >
> > > ### Re Precise Statement of Lemma 2.
> > >
> > > Due to word limitations, we were not able to show a precise description of Lemma 2 in the rebuttal. Now we show the exact description as:
> > >
> > > Given an $n$ layer network, let the parameters of the random weights after mapping $n-r$ times be $\mu^\prime$ and $\sigma^\prime$, $x^{l}$ denotes the output feature map of layer $l$ and $W^{l}$ denotes the weights in layer $l$. The output difference $Y_1-Y_2$ is hoped to be smaller than a minimal term $\epsilon_y$. The probability of $(Y_1-Y_2) < \epsilon_y$ is defined as $P((Y_1-Y_2) < \epsilon_y)$. And $P((Y_1-Y_2) < \epsilon_y)$ is expected to be larger than $ F ( \sqrt{2}\beta / 2 )$. To meet this constrain, we have $ P((Y_1-Y_2) < \epsilon_y) \geqslant F ( \sqrt{2}\beta / 2 )$, $\sum_{i = 1}^m \sigma_i^\prime$ is upper bounded as,
> > >
> > > $\sum_{i = 1}^m \sigma_i^\prime \leqslant \frac{\epsilon_y}{x^{n-1} \cdot W^{n}\beta}$，
> > >
> > > where $m$ is the size of the feature map.
> > >
> > > ### Re Notation.
> > >
> > > Thanks for giving a more rigorous expression. The notation will be changed into the standard format as is mentioned.

---

> > > > ### Comment · Reviewer_1Bpz · 2023-08-14
> > > > **Lemma 1**
> > > >
> > > > Can you explain please why you claim that the conditions: $\epsilon_r'=\sum_i\mu_i + \delta$  and $\delta\to 0$ imply that $\epsilon\to 0$? It only implies that $\epsilon_r'\to\sum_i\mu_i>0$. That choice of $\epsilon$ does not violate any assumptions in the lemma. Since $\epsilon$ is an arbitrary constant chosen in the guarantee statement, not something learned during training, it can be chosen independently of the value of $\sum_i\mu_i$. Suppose that $\sum_i\mu_i=986.3$ as you mention in your response. Then, I can choose $\epsilon_r'=\sum_i\mu_i + 10^{-100}$, which means that as long as $\sum_i\sigma_i\gg 10^{-100}$, the cosine similarity will be close to zero according to the lemma, but this does not seem right because the cosine similarity will be close to 1 in that case.

---

> > > > > ### Author Response · Authors · 2023-08-15
> > > > > **Official Comment by Authors about Lemma 1**
> > > > >
> > > > > Thanks for the response. We further clarify the condition as follows:
> > > > >
> > > > > We provided the clear definition of $\epsilon_r^\prime$ in the Lemma 1 as
> > > > >
> > > > > $\epsilon_r^\prime= \epsilon_r \cdot \|W_r[1]\|_p \|W_r[2]\|_p$.
> > > > >
> > > > > In the paper, $\epsilon_r$ is a constant close to 0 as is assumed both in Lemma 1 at line 143 and before the lemma at line 140. Since $\epsilon_r$ is close to 0, $\epsilon_r^\prime$ is also close to 0, as is mentioned in Lemma 1. Thus, given the large value of $\sum^m_{i=1} \mu_i$, we cannot choose $\epsilon_r^\prime = \sum^m_{i=1} \mu_i + 10^{-100}$.
> > > > >
> > > > > The reason why we assume that $\epsilon_r$ is close to 0 is that the objective of Lemma 1 is to constrain the cosine similarity. If we choose $\epsilon_r > 1$, then $P(cos(W^{r}[1], W^{r}[2]) \leqslant \epsilon_r)>F(\alpha)$ always holds true, since $P(cos(W^{r}[1], W^{r}[2]) \leqslant \epsilon_r)=1$ and $F(\alpha) < 1$. In this case, Lemma 1 still stands even if the cosine similarity is close to 1. But it is out of scope of this paper since we only focus on the scenarios of $\epsilon_r \to 0$. Thus, in this paper, we assume $\epsilon_r \to 0$ in Lemma 1.

---

> > > > > > ### Author Response · Authors · 2023-08-17
> > > > > > **Further Clarification**
> > > > > >
> > > > > > In order to clearly justify the two Lemmas in this paper, we provide a polished and more precise Lemma 1 and 2 here.
> > > > > >
> > > > > > &nbsp;
> > > > > >
> > > > > > ### Lemma 1
> > > > > >
> > > > > > Consider a fixed $W^{r}[1]$ with size of $m$, and a constant $\epsilon_r$ that satisfies $\epsilon_r = o(1)$. Let $\epsilon_{r}^\prime = \epsilon^r \times \|Wr[1]\|_p \|Wr[2]\|_p$,
> > > > > >
> > > > > > we have $\epsilon_r^\prime = o(1)$. Assume that $\sum_{i = 1}^{m}\mu_i \gg 0$. Let $F$ denote the cumulative function of a standard Gaussian distribution, the constraint $cos(W^{r}[1], W^{r}[2]) \leqslant \epsilon_{r}$ holds with the probability at least $F(\alpha)$, i.e. $P(cos(W^{r}[1], W^{r}[2]) \leqslant \epsilon_{r}) > F(\alpha)$, if
> > > > > >
> > > > > > $\sum_{i = 1}^{m}\sigma_i \geqslant \frac{\epsilon_{r}^\prime - \sum_{i = 1}^{m}\mu_i}{\alpha\sum_{i = 1} ^{m}|W^r[1]_i|} > 0$,
> > > > > >
> > > > > > where $m$ is the size of $W^r$, and for any variation $a$, $a_i$ denotes the $i$th element of $a$.
> > > > > >
> > > > > > &nbsp;
> > > > > >
> > > > > > ### Lemma 2
> > > > > >
> > > > > > Given an $n$ layer network and assuming $ReLU$ to be the activation function, let the parameters of the random weights after mapping $n-r$ times be $\mu^\prime$ and $\sigma^\prime$, $x^{l}$ denote the output feature map of layer $l$ and $W^{l}$ denote the weights in layer $l$. The output difference $Y_1-Y_2$ is hoped to be smaller than a minimal term $\epsilon_y$. The probability of $(Y_1-Y_2) < \epsilon_y$ is defined as $P((Y_1-Y_2) < \epsilon_y)$. And $P((Y_1-Y_2) < \epsilon_y)$ is larger than $ F ( \frac{\sqrt{2}\beta}{2} )$, i.e. $ P((Y_1-Y_2) < \epsilon_y) \geqslant F ( \frac{\sqrt{2}\beta}{2} )$, if $\sum_{i = 1}^m \sigma_i^\prime$ is upper bounded as,
> > > > > >
> > > > > > $\sum_{i = 1}^m \sigma_i^\prime \leqslant \frac{\epsilon_y}{x^{n-1} \cdot W^{n}\beta}$，
> > > > > >
> > > > > > where $m$ is the size of the feature map.
> > > > > >
> > > > > > &nbsp;
> > > > > >
> > > > > > We are looking forward to your response.

---

> > > > > > > ### Comment · Reviewer_1Bpz · 2023-08-18
> > > > > > > **Lemma 1**
> > > > > > >
> > > > > > > Unfortunately, this doesn't fix the issues. For example, you now assume that $\epsilon_r'=o(1)$ and $\sum\mu_i\gg 0$ and then require in the bound that the difference $\epsilon_r'-\sum\mu_i>0$. These assumptions contradict each other.

---

> > > > > > > > ### Author Response · Authors · 2023-08-19
> > > > > > > > **Official Comment by Authors**
> > > > > > > >
> > > > > > > > Thanks for the response. Given $\epsilon_r^\prime = o(1)$ and $\sum_{i = 1}^{m}\mu_i \gg 0$, we have $\epsilon_{r}^\prime - \sum_{i = 1}^{m}\mu_i < 0$. In our bound, we have $\frac{\epsilon_{r}^\prime - \sum_{i = 1}^{m}\mu_i}{\alpha\sum_{i = 1} ^{m}|W^r[1]_i|} > 0$.
> > > > > > > >
> > > > > > > > Only if $\alpha > 0$ would those assumptions contradict. But in fact, $\alpha$ is negative in our proof, as mentioned in Line 30 in our supplementary material. Thus, $\frac{\epsilon_{r}^\prime - \sum_{i = 1}^{m}\mu_i}{\alpha\sum_{i = 1} ^{m}|W^r[1]_i|} > 0$ holds.

---

> > > > > > > > > ### Comment · Reviewer_1Bpz · 2023-08-20
> > > > > > > > > **Response**
> > > > > > > > >
> > > > > > > > > Line 30 in the supplementary material is not a proof that $\alpha<0$ as far as I can tell. It's basically an acknowledgement that the bound leads to a contradiction when $\alpha>0$. Does this mean that $\alpha<0$? No, because $\alpha$ is a free parameter we choose according to the level of confidence we desire. Besides, since $F(\alpha)$ is the standard Gaussian CDF, doesn't $\alpha<0$ imply a probability of less than 1/2, meaning it's a low confidence guarantee?

---

> > > > > > > > > > ### Author Response · Authors · 2023-08-21
> > > > > > > > > > **Official Comment by Authors**
> > > > > > > > > >
> > > > > > > > > > Thanks for the response. We agree that $\alpha < 0$ indicates a low confidence guarantee in Lemma 1 and we should include the condition that $\alpha < 0$ explicitly in Lemma 1. Thanks for pointing it out. We will revise Lemma 1 accordingly. However, we argue that the low confidence guarantee does not indicate the ineffectiveness of Lemma 1. Although we focus on the scenario where $F(\alpha) < 0.5$, this bound still plays an important role in our algorithm.
> > > > > > > > > >
> > > > > > > > > > In fact, without the lower bound constraint in Lemma 1, the probability $P(cos(W^{r}[1], W^{r}[2]) \leqslant \epsilon_{r})$ becomes $0$ in practice. To illustrate it, we report the number of times which satisfies the condition that $P(cos(W^{r}[1], W^{r}[2]) \leqslant \epsilon_{r})$ after sampling $1.0 \times 10^4$ for ResNet-18 on CIFAR-10/100 with/without constraints, as shown in the following table:
> > > > > > > > > >
> > > > > > > > > > | Dataset  | with/without constrain    | Total examples | Number of examples satisfies cos < $\epsilon$ | Ratio of examples satisfies cos < $\epsilon$ | Minimum of cosine similarity | Robust accuracy under $PGD^{20}$ |
> > > > > > > > > > |----------|------------|----------------|-----------------------------------------------|----------------------------------------------|------------------------------|----------------------------------|
> > > > > > > > > > | CIFAR10  | without  | 10000          | 4558                                          | 0.4558                                       | -0.0557                      | 69.48                            |
> > > > > > > > > > | CIFAR10  | with(ours) | 10000          | 0                                             | 0                                            | 0.7835                       | 54.59                            |
> > > > > > > > > > | CIFAR100 | without  | 10000          | 4124                                          | 0.4124                                       | -0.0669                      | 42.01                            |
> > > > > > > > > > | CIFAR100 | with(ours) | 10000          | 0                                             | 0                                            | 0.7361                       | 30.05                            |
> > > > > > > > > >
> > > > > > > > > > As shown in the table,  $P(cos(W^{r}[1], W^{r}[2]) \leqslant \epsilon_{r})$ becomes $0$ without constraints, which indicates that it is difficult for random weights to have a smaller cosine similarity if $\sigma$ is directly involved in the optimization without a lower bound. However, with the involvement of Lemma1, this probability becomes $0.4558$ on CIFAR10 and $0.4124$ on CIFAR100. Although $P < 0.5$ can be regarded as a low confidence guarantee, it is sufficient for our algorithm to achieve state-of-the-art performance compared with other baselines.
> > > > > > > > > >
> > > > > > > > > > In summary, Lemma1 is meaningful for guiding the design of the algorithm in this paper. Further, it can be seen that the expectations of Lemma1 on robustness match the experimental results

---

### Official Review · Reviewer_8U3B · 2023-06-29

**Soundness:** 3 good
**Presentation:** 3 good
**Contribution:** 3 good
**Rating:** 7
**Confidence:** 4

**Summary:**

This paper proposes a random injection approach to improve the adversarial robustness against attacks. Different from previous work, the proposed algorithm includes random weights in the optimization and imposes constraints for better trade-offs. The constraints rely on proving the following Lemma:

1.(Lemma 1). The variance of distribution where the noise is sampled is lower bounded to ensure lower gradient similarity after sampling.

2.(Lemma 2). The variance of distribution is also upper bounded to ensure lower natural classification error.

The proposed algorithm combines the lower and upper bound to perform adversarial training with random weights under constraints. Several experiments are provided to verify the effectiveness of the proposed algorithm.


**Strengths:**

1.The theoretical analysis provides good insight that leads to the proposed constraints, which could benefit randomized defense community.

2.The evaluation is conducted on various datasets and models, and the proposed algorithm achieves reasonable improvement over baselines.


**Weaknesses:**

1.Lack of assumption before or in the lemma, which makes it difficult to verify the scope.

2.Although the proposed algorithm proposes to perform adversarial training in a black-box manner, there is no evaluation of black-box attacks in the experiment section. The authors should conduct more experiments to verify the performance of proposed algorithm under popular black-box attacks.


**Questions:**

I think more clarification and evaluation are needed for the concerns mentioned in the Weakness section. I would appreciate the authors making amends along those directions.

**Limitations:**

The evaluation of proposed algorithm under black-box attacks is relatively limited.

---

> ### Author Rebuttal · Authors · 2023-08-08
>
> ### Re Lack of assumption.
> Thanks for the suggestions on the lack of rigor in our theory. Lemma2 inference requires that the nonlinear layer maintains the conditions mentioned in Equation 14 in the supplementary material. In addition the lower bound of $\sigma$ should be greater than 0 to ensure the randomness of the random weights. We will add this section to the text in the final version to ensure that the theory is as rigorous as possible.
>
> ### Re Lack of black-box attack.
> Thanks for the suggestion. We include more evaluation of black-box attacks, such as Square and Pixle. We conduct evaluation ResNet-18 on CIFAR-10. The results are shown in the following Table:
>
> | Method     | Square | Pixle |
> |------------|--------|-------|
> | baseline   | 54.68  | 8.10  |
> | CTRW(ours) | 77.73  | 72.14 |
>
>
> As shown in the table, our proposed algorithm achieves better performance under black-box attacks, especially under Pixle attack.

---

> > ### Author Response · Authors · 2023-08-13
> >
> > Thank you again for your comments and suggestions. We would like to know if we have addressed your issues. Meanwhile, if you have any other concerns, we are open to further discussion.

---

> > > ### Comment · Reviewer_8U3B · 2023-08-15
> > > **Post-rebuttal comments.**
> > >
> > > Thanks for authors addressing my concerns, which solved my questions.
> > >
> > > I have also read the other reviews and corresponding feedback.
> > >
> > > In my opinion, I think this is a good paper and I vote for acceptance.
> > >
> > > Therefore, I increase my score by one.

---

> > > > ### Author Response · Authors · 2023-08-17
> > > > **Official Comment by Authors**
> > > >
> > > > Thanks for providing suggestions and support. We have revised an accurate Lemma based on the suggestions given by you. We sincerely value the suggestions given by you as it helps a lot in the rigor of this paper. Please feel free to interact with us for any further questions.

---

### Official Review · Reviewer_kSR2 · 2023-07-02

**Soundness:** 3 good
**Presentation:** 3 good
**Contribution:** 3 good
**Rating:** 8
**Confidence:** 4

**Summary:**

The authors attempt to analyse the effect of the design of noise in random networks on the network and to improve the network adversarial robustness.

The authors suggest designing an interval to constrain the intensity of the noise within a desirable range.

The authors conducted experiments based on CNNs and the results support these conclusions.

**Strengths:**

S1) The paper attempts to answer the question of the relationship between adversarial robustness of stochastic networks and noise. This is one of the topics of the NeurIPS audience.

S2) This paper provides a mathematical analysis that explains, from a theoretical point of view, the effect of random noise strength on network performance. The results of the analysis are also interesting: an interval is used to constrain the noise to achieve relatively desirable results.

S3) This work shows that the optimisation of noise in random networks has a large impact on network adversarial robustness. Their success also provides an interesting way of thinking about the problem of random network design.

S3) The results are interesting and intuitive, and consistent with the theoretical analysis.

**Weaknesses:**

W1) Lack of code usability statements, although the network structure is relatively simple.

W2) Lack of evaluation under non-adversarial training may be a potential weakness, but existing evaluations are sufficiently representative.


**Questions:**

Please see Weakness part

**Limitations:**

This is a solid paper that presents a complete analysis and interesting results. The paper provides novel ideas for random network design. As an improvement, the links to the code is needed for more adequate support to the data in the paper.

---

> ### Author Rebuttal · Authors · 2023-08-09
>
> ### Re Code usability.
> Thanks for this nice concern about code usability. According to this year's rules, the code about our method has been given to Area Chairs in the form of an anonymous link.
>
> ### Re Lack of experiment.
> Thanks for this nice concern. It is interesting to see the performance without adversarial training. We conduct evaluation with ResNet-18 on CIFAR-10. All the algorithms are trained with natural training. The results are shown in the following table:
>
> | Method     | Natural | $PGD^{20}$ |
> |------------|---------|------------|
> | baseline   | 84.17   | 0.00       |
> | CTRW(ours) | 84.61   | 3.44       |
>
>
>
> As shown in the table, our proposed algorithm can achieve better adversarial robustness than the baselines, which demonstrates the effectiveness of our algorithm. Compared with the results after adversarial training, the results without adversarial training cannot be competitive. Thus, adversarial training is still a necessary training strategy in our algorithm.

---

> > ### Author Response · Authors · 2023-08-13
> >
> > Thank you again for your comments and suggestions. We would like to know if we have addressed your issues. Meanwhile, if you have any other concerns, we are open to further discussion.

---

> > > ### Comment · Reviewer_kSR2 · 2023-08-15
> > > **Post-rebuttal**
> > >
> > > The authors have addressed my major concerns about the submission, so I would like to recommend an acceptance to the paper.

---

> > > > ### Author Response · Authors · 2023-08-17
> > > > **Official Comment by Authors**
> > > >
> > > > Thank you for the advice and support you have provided. We sincerely appreciate your addition to the experiments in this paper. This has helped us to more accurately define the scope of this work. We will critically claim the scope of use of the methods presented in this paper based on your suggestions. Please feel free to interact with us for any further questions.

---

### Official Review · Reviewer_nrzD · 2023-07-19

**Soundness:** 4 excellent
**Presentation:** 4 excellent
**Contribution:** 4 excellent
**Rating:** 6
**Confidence:** 2

**Summary:**

This paper proposes to incorporate random weights into the optimization to exploit the potential of randomized defense. A theoretical analysis of the connections between randomness parameters and gradient similarity as well as natural performance is also provided. The method is evaluated on several datasets and benchmark CNNs to verify the utility.

**Strengths:**

The paper addresses a novel task and presents a unique method to handle it.

The paper is well written and easy to follow.

Theoretical analysis is provided in this paper which can provide more insights in the field.

The reported results are impressive.

**Weaknesses:**

The method is only evaluated on CNN-based architectures. Since transformer has been widely adopted on both vision and language tasks, it  will be helpful to provide the results on transformer-based architectures.

From Figure 3(c), we can see that for resnet18,  with the increasing of the PGD attack steps, the robust accuracy first quickly decreases and then  increases slowly after about 10 steps, can you provide some explanation about this phenomenon?

The authors state that: to verify the accidental of the results, the authors repeated the evaluation on CIFAR-10 multiple times, and the results are illustrated in Table 5. However, how many times the experiments are repeated should be explicitly presented in the paper.

**Questions:**

More analysis should be provided for the curves of the robust accuracy for resnet18 in Figure 3(c).

The repeated number of the experiments illustrated in Table 5 should be explicitly presented.

**Limitations:**

Yes

---

> ### Author Rebuttal · Authors · 2023-08-09
>
> ### Re Experiments on ViT.
>
> Thanks for the suggestions regarding the experimental integrity of this work. Our proposed algorithm can be simply deployed to other neural networks, such as ViTs. For illustration, we deploy CTRW on ViT-S and evaluate the performance on CIFAR-10. The results are shown in the following table:
>
> | Method     | cw20  | $PGD^{20}$ |
> |------------|-------|------------|
> | baseline   | 34.62 | 33.49      |
> | CTRW(ours) | 45.21 | 45.68      |
>
> As can be seen in these results, our proposed algorithm can be easily adopted by different neural networks. On ViTs, our proposed algorithm achieves better adversarial robustness than the baseline.
>
> ### Re Trends in Robust Accuracy.
> The trend of robust accuracy as shown in Fig. 3 is a constant trend in the method of designing path migration. The method designed in this paper actually makes a difference between the paths at attack time and at inference time. When the number of PGD steps is small, the gradient rise is not sufficient. At this point, as the number of PGD steps increases, the gradient rises along a path that is very close to the direction of the optimization path, which has a greater impact on the network. Therefore the adversarial robustness gradually decreases. However, when the number of PGD steps is high, the gradient rise is already sufficient, and at this time, as the gradient rises, the difference in direction of the gradient at each step is getting larger and larger compared with the gradient at the optimization corresponding to it. So the attack becomes less and less effective.
>
> ### Re Times of the Experiments.
> Thanks for this nice concern. All experiments in this paper were repeated 10 times and averaged. We will clearly state it in the final version.

---

> > ### Author Response · Authors · 2023-08-13
> >
> > Thank you again for your comments and suggestions. We would like to know if we have addressed your issues. Meanwhile, if you have any other concerns, we are open to further discussion.

---

### Decision · Program_Chairs · 2023-09-21

**Decision:**

Accept (poster)

**Comment:**

This work proposes an efficient Black-box Adversarial Training method called Constrained Trainable Random Weight (CTRW). It incorporates random weight parameters into the optimization process and includes theoretically-guided bounds to ensure better trade-offs. The reviewers acknowledge the valuable insights provided by the theoretical analysis. However, they also raise concerns about experimental limitations (Reviewer nrzD, kSR2, 8U3B), as well as concerns regarding a lemma (Reviewer 8U3B, 1Bpz) and novelty (Reviewer 1Bpz). Following the rebuttal, the concerns raised by Reviewers 8U3B and kSR2 have been addressed, resulting in an increased rating of "Accept" and "Strong Accept," respectively. However, Reviewer 1Bpz still expresses concerns about the correctness of Lemma 1 and Lemma 2 even after the rebuttal and maintains his/her rejection score.

One of the main concerns raised by Reviewer 1Bpz is regarding Lemma 1. The lemma holds when $\epsilon_r'-\sum_i \mu_i+\delta$ for any $\delta>0$. However, as $\delta\rightarrow 0^+$, the condition is satisfied by choosing $\sigma\rightarrow 0^+$, which implies that the cosine similarity approaches 1. The authors argue that in practice, $\delta\rightarrow 0^+$ does not exist since $\epsilon_r'\rightarrow 0$ and $\sum_i \mu_i \nrightarrow 0$. In response, the reviewer points out the contradiction of $\epsilon_r'-\sum_i \mu_i > 0$ since $\sum_i \mu_i$ can always be positive. Then, the authors mention that $\alpha$ is negative, but the reviewer insists that $\alpha$ is freely chosen based on the desired level of confidence. Ultimately, the authors state that their theory can only be applied when $\alpha < 0$, meaning confidence is less than 0.5.

There are two main limitations for the theory: 1) Lemma 1 only studies the probability of $cos(W1, W2) < \epsilon_r$ where $\epsilon_r\rightarrow 0^+$, i.e., $W1$ and $W2$ being completely different. The assumption $\epsilon_r\rightarrow 0$ is too strong; 2) The theory is only valid under the condition of $\alpha < 0$, which is not sufficiently general.

On the other hand, these lemmas also have positive aspects, as noted by the reviewer: 1) This paper attempts to answer the question of the relationship between the adversarial robustness of stochastic networks and noise, which is interesting and warrants further exploration; 2) The results of the theoretical analysis are also interesting: an interval is used to constrain the noise to achieve relatively desirable results. Based on the high praise from Reviewers nrzD, kSR2, and 8U3B regarding the theoretical findings (resulting in weak accept, accept, and strong accept scores), as well as the practical application of these derived results in Adversarial Training, which has led to significant improvements, I lean towards acceptance. I also suggest that the authors revise the paper according to the suggestions of the reviewers and further justify Lemma 1 and Lemma 2 in the final version.